# FLAIR: a Country-Scale Land Cover Semantic Segmentation Dataset From Multi-Source Optical Imagery

**Anatol Garioud**[1]   **Nicolas Gonthier**[2]   **Loic Landrieu**[2,3]   **Apolline De Wit**[1]
**Marion Valette**[1]   **Marc Poupée**[4]   **Sébastien Giordano**[1]   **Boris Wattrelos**[1]

[1]French National Institute of Geographical and Forest Information (IGN)
[2]Univ Gustave Eiffel, IGN, ENSG, LASTIG
[3]LIGM, Ecole des Ponts, Univ Gustave Eiffel, CNRS
[4]National School of Geographic Sciences (ENSG)
`{firstname.lastname}@ign.fr`

## Abstract

We introduce the French Land cover from Aerospace ImageRy (FLAIR), an extensive dataset from the French National Institute of Geographical and Forest Information (IGN) that provides a unique and rich resource for large-scale geospatial analysis. FLAIR contains high-resolution aerial imagery with a ground sample distance of 20 cm and over 20 billion individually labeled pixels for precise land-cover classification. The dataset also integrates temporal and spectral data from optical satellite time series. FLAIR thus combines data with varying spatial, spectral, and temporal resolutions across over $817 \text{ km}^2$ of acquisitions representing the full landscape diversity of France. This diversity makes FLAIR a valuable resource for the development and evaluation of novel methods for large-scale land-cover semantic segmentation and raises significant challenges in terms of computer vision, data fusion, and geospatial analysis. We also provide powerful uni- and multi-sensor baseline models that can be employed to assess algorithm's performance and for downstream applications. Through its extent and the quality of its annotation, FLAIR aims to spur improvements in monitoring and understanding key anthropogenic development indicators such as urban growth, deforestation, and soil artificialization. Dataset and codes can be accessed at https://ignf.github.io/FLAIR/

## 1   Context

According to a 2015 report by the Food and Agriculture Organization of the United Nations (FAO) [1], approximately 75% of the world's soils are in fair, poor, or very poor condition. This degradation poses significant threats to the health and long-term sustainability of ecosystems. Healthy soils provide invaluable ecosystem services, including: (i) providing natural habitats for numerous plant and animal species [2], (ii) acting as the largest carbon sink, surpassing the atmosphere and all combined biomass [3], and (iii) functioning as a rainwater reservoir, supporting food production and storing freshwater [4].

The degradation of soils and biodiversity is largely attributed to *land artificialization* [1], which causes long-term damage to the biological, hydrological, climatic, and agronomic functions of the soil due to its occupation or use [5, 6]. In order to effectively monitor and manage land artificialization, public authorities have expressed the need for scalable land-cover monitoring tools. With the increasing availability of high-quality Earth Observation (EO) data, the French National Institute

37th Conference on Neural Information Processing Systems (NeurIPS 2023) Track on Datasets and Benchmarks.

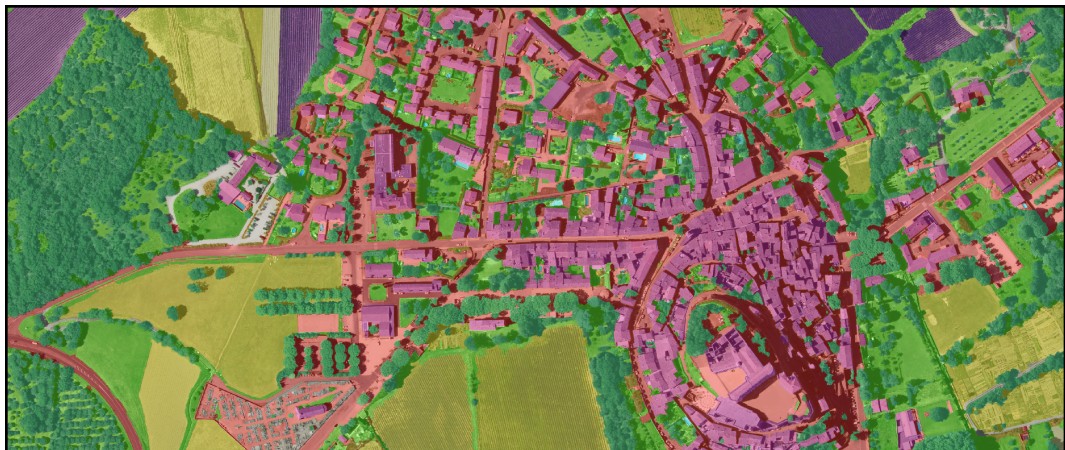

**Figure 1: Detail from the FLAIR Dataset.** The very high resolution annotation of 20 cm allows us to distinguish the exact extent of individual houses, roads, and trees.

of Geographical and Forest Information (IGN) [7] is exploring the use of artificial intelligence to automatically conduct high-resolution, global-scale land-cover mapping, an essential component for addressing soil degradation at a national level [8]. A central aspect of this initiative is to create and disseminate precise and up-to-date reference datasets for researchers and policy-makers.

We introduce the French Land cover from Aerospace ImageRy (FLAIR) dataset, the largest multi-sensor land-cover dataset with very-high-resolution annotations. FLAIR combines very-high-resolution (VHR, 20cm) images, photogrammetry-derived surface models, and optical Sentinel-2 multi-spectral satellite time series with a nominal revisit time of 5 days at the equator. The diverse spatial, spectral, and temporal resolutions of these acquisitions offer valuable complementary perspectives for land cover analysis. Over 20 billion pixels have been hand-annotated by geospatial experts, using a nomenclature of 19 land-cover classes. The data spans $817 \text{ km}^2$ across 50 French sub-regions featuring diverse bioclimatic attributes at various times of the year, thus displaying complex and challenging domain shifts.

FLAIR combines data sources with heterogeneous spatial, temporal, and spectral resolutions and high-precision annotations, and aims to foster the development of new large-scale semantic segmentation methods. Given its scale and the complexity of the domain shifts it exhibits, FLAIR also presents an exciting challenge for the computer vision and machine learning communities.

## 2 Related Work

Numerous land-cover datasets have been introduced to train semantic segmentation methods, see Table 1. Existing datasets usually present a trade-off: they either offer high-resolution annotations but cover a small extent (like Vaihingen [9]), or provide large-extent coverage but with low-resolution annotations (such as BigEarthNet [10] or SEN12MS [11]). In contrast, FLAIR offers both very high-resolution annotations (20 cm) while covering a large portion of the French territory.

FLAIR comprises over 20 billion individually, manually annotated pixels, which is over 1000 times more than SEN12MS and 2 times more than BigEarthNet, which employs semi-automatic annotations. The DeepGlobe and LoveDA datasets, the closest counterparts to our dataset, provide a large coverage of $1717 \text{ km}^2$ at 50 cm and $536 \text{ km}^2$ at 30 cm respectively. However, FLAIR provides over 3 times as many annotated pixels and a higher resolution.

The spatial resolution of the annotation is crucial in land-cover analysis. Insufficient resolution prevents the precise measurements of surfaces and boundaries. Furthermore, small-scale features, such as individual houses, lone trees or roads, may not be captured accurately, limiting the potential applications of the derived segmentation.

**Table 1: Land Cover Datasets.** Publicly available datasets for semantic segmentation of land cover from remote sensing Earth observation imagery.

| Dataset | Annotation | | | | Acquisition | | |
|---|---|---|---|---|---|---|---|
| | Pixels $\times 10^6$ | Resolution | Classes | Source | Resolution | Extent (km$^2$) | Source |
| SAT-4/SAT-6 [12] | 0.9 | 28 m | 4/6 | semi-automatic (NLCD [13]) | 1 m | 13 860k | aerial |
| SEN12MS [11] | 14 | 100 m | 17 | fully-automatic (MODIS [14]) | 10 m | 3 551k | Sentinel-1&-2 |
| Vaihingen [9] | 82 | 8 cm | 6 | visual interpretation | 8 cm | 1 | aerial |
| EuroSAT [15] | 110 | 50 m | 10 | EU Urban Atlas [16] | 10 m | 11 059 | Sentinel-2 |
| MultiSenGE [17] | 534 | 10 m | 14 | visual interpretation | 10 m | 57 433 | Sentinel-1&-2 |
| Landcovernet [18] | 589 | 10 m | 7 | semi-automatic (MODIS [14]) | 10 m | 58 982 | Sentinel-2 |
| MiniFrance [19] | 1 510 | 50 m | 14 | EU Urban Atlas [16] | 50 cm | 53 000 | aerial |
| DynamicEarthNet [20] | 1 889 | 3 m | 7 | visual interpretation | 3 m | 16 986 | Sentinel-1&-2, PlanetFusion |
| OpenEarthMap [21] | 4 931 | 25–50 cm | 8 | visual interpretation | 25–50 cm | 799 | aerial, UAV, satellite |
| Five-Billion-Pixels [22] | 5 000 | 4 m | 24 | visual interpretation | 4 m | 50 000 | Gaofen-2 |
| LoveDA [23] | 6 000 | 30 cm | 7 | visual interpretation | 30 cm | 536 | aerial |
| DeepGlobe [24] | 6 867 | 50 cm | 7 | visual interpretation | 50 cm | 1 717 | Wordlview-2/3, GeoEye-1 |
| BigEarthNet [10] | 8 500 | 100 m | 19 | semi-automatic (CLC [25]) | 10 m | 850 k | Sentinel-1&-2 |
| **FLAIR** | **20 385** | **20cm** | **19** | **visual interpretation** | **20 cm/10 m** | **817** | **aerial, Sentinel-2** |

# 3  Dataset Description

FLAIR combines granular pixel annotation with heterogeneous data sources across a large and diverse spatio-temporal extent.

## 3.1  Extent & Annotation

**Spatio-Temporal Distribution.** The FLAIR dataset consists of 77 762 patches represented in Figure 3. Each patch includes a high-resolution aerial image of 0.2 m, a yearly satellite image time series with a spatial resolution of 10 m, and pixel-precise elevation and land cover annotations at 0.2 m resolution. As shown in Figure 5, the acquisitions are taken from 916 unique areas distributed across 50 French spatial domains (*départements*), covering approximately 817 km$^2$. Aerial images were captured under favorable weather conditions between April and November from 2018 to 2021. Each satellite time series corresponds to the entire year of acquisition of the matching aerial image.

**Annotations.** Each pixel has been manually annotated by photo-interpretation of the 20 cm resolution aerial imagery, carried out by a team supervised by geography experts from the IGN. During the annotation process, we initially identified 18 classes. We group certain classes together due to the rarity of certain classes, such as swimming pool, greenhouse, or snow, or potential ambiguity, as seen with ligneous and mixed vegetation. The resulting 12-class nomenclature leads to more statistically robust evaluation metrics. Nonetheless, users can still access and use the extended nomenclature.

**Table 2: Land-cover Class Distribution.** Semantic nomenclature used by the FLAIR dataset and their proportion among the entire dataset.

| Class | % | Class | % | Class | % |
|---|---|---|---|---|---|
| (1) building | 7.1 | (6) coniferous | 4.3 | (11) agricultural land | 12.8 |
| (2) pervious surface | 7.3 | (7) deciduous | 17.3 | (12) plowed land | 3.5 |
| (3) impervious surface | 12.1 | (8) brushwood | 6.3 | (13) other | 0.8 |
| (4) bare soil | 3.1 | (9) vineyard | 3.0 | | |
| (5) water | 4.5 | (10) herbaceous vegetation | 18.2 | | |

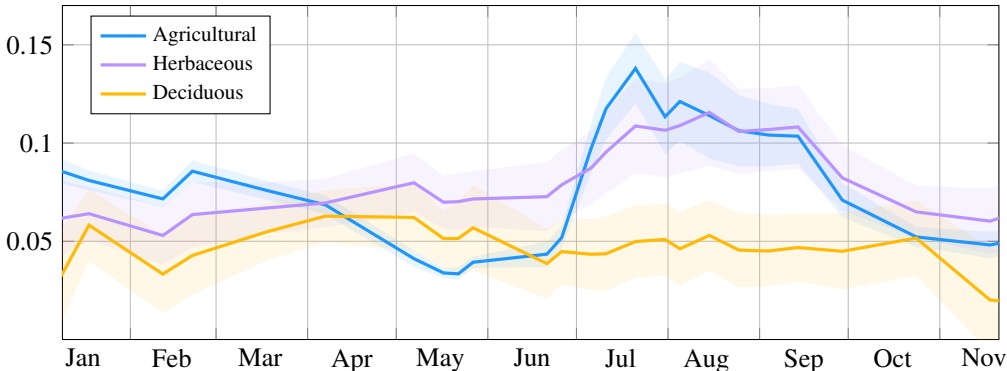

**Figure 2: Land Cover Spectral Dynamics.** We represent the temporal progression of reflectance values for the red channel (B2) for different land cover types using Sentinel-2 satellite data. Each curve represents the mean reflectance for a land cover types over a subset of the Ardèche department in 2020, with shaded regions indicating the standard deviation. This plot illustrates the distinct spectral dynamics of different land cover types.

Movable objects like cars or boats are annotated according to their underlying cover. Table 2 outlines the class set and their distribution. Refer to the appendix for more details.

Thanks to the high resolution of the aerial images, anthropic structures like roads and buildings can be identified with a high level of detail. Agricultural and natural lands like forests or herbaceous cover, which make up over 65% of the dataset, can often be challenging to distinguish from images alone. As shown in Figure 2, multispectral satellite time series prove to be particularly effective in characterizing the temporal evolution of plant phenology [26, 27], a key motivation for incorporating these into the dataset.

**Training Splits.** The dataset is made up of 50 distinct spatial domains, aligned with the administrative boundaries of the French *départements*. For our experiments, we designate 32 domains for training, 8 for validation, and reserve 10 as the official test set (refer to Figure 5 or appendix). This arrangement ensures a balanced distribution of semantic classes, radiometric attributes, bioclimatic conditions, and acquisition times across each set. Consequently, every split accurately reflects the landscape diversity inherent to metropolitan France. It is important to mention that the patches come with meta-data permitting alternative splitting schemes, for example focused on domain shifts.

## 3.2 Acquisitions

FLAIR offers 3 complementary sources of acquisition, each with distinct nature and spatial/spectral/temporal resolutions: aerial images, elevation models, and satellite image time series.

**Very High Resolution Aerial Images.** The aerial images are taken from IGN's free license BD-Ortho product [28]. All aerial images are $512 \times 512$ in size with a resolution of $20\,\text{cm}$ per pixel, and feature 4 spectral channels: red, blue, green, and near-infrared. Each patch comes with metadata such as the date and time of acquisition, geographical location and altitude of the patch centroid, and specifics about the camera used for acquisition. All images are aligned to a shared cartographic coordinate reference system (EPSG:2154), and radiometric corrections are applied to ensure homogeneity per spatial domain [29]. This means that colors should not be interpreted as physical measurements of channel reflectance.

**Elevation.** Each aerial image is accompanied by an elevation value produced by the IGN. This information is not an independent measurement but a product derived from a digital elevation model and a digital surface model obtained through photogrammetry on the aerial images, thereby ensuring temporal consistency.

**Sentinel-2 Time Series.** Each patch is associated with a satellite image time series from the Sentinel-2 constellation [30], as shown in Figure 4. Each image in the sequence is of size $40 \times 40$ with a 10 m pixel resolution, centered around the aerial image. Each pixel is characterized by 10

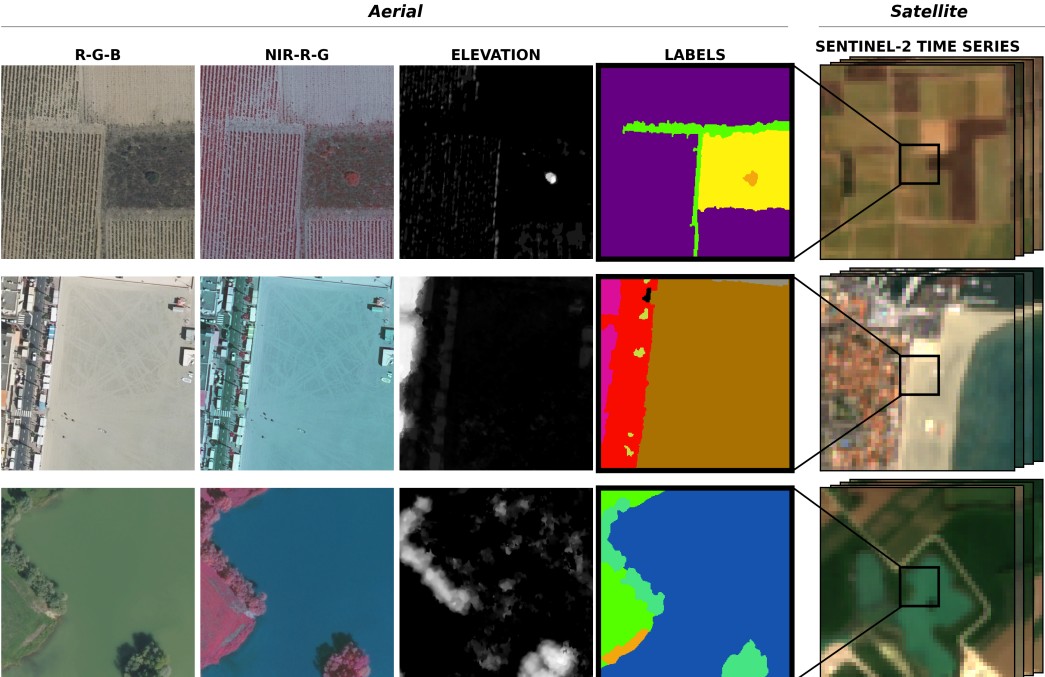

**Figure 3: Patches from FLAIR.** The dataset is comprised of 77 762 patches. Each patch contains (i) a $512 \times 512$ aerial image at 0.2m resolution with red, green, blue (RGB) and near-infrared (NIR) values, (ii) a pixel-precise digital surface model providing an elevation for each pixel, (iii) semantic labels for each pixel, and (iv) an optical time series of spatial dimension $40 \times 40$ and 10m per pixel, centered on the aerial image.

spectral bands, ranging from the visible to the medium infrared spectrum; additional details can be found in the appendix. The time series span the entire year during which the corresponding aerial image was acquired and contain from 20 to 110 images, depending on the satellite availability and the orbit characteristics. We include acquisitions with cloud cover and provide cloud and snow probability masks obtained with Sen2cor [31] in the metadata, alongside information about the satellite and its orbit.

Only the spatial and temporal extents of the aerial images are annotated. Terrain features that evolve over time, such as changing river banks or tidal patterns, may not be consistent throughout the time series. Nevertheless, the satellite time series provide invaluable spectral and temporal information, as well as a broader spatial context of $400 \times 400$m.

### 3.3 Specificities

The FLAIR dataset presents specificities often encountered in geospatial analysis but seldom in computer vision: large-scale multi-sensor acquisitions, and complex spatio-temporal domain shifts.

**Multi-Sensor.** FLAIR combines optical acquisitions with drastically different spatial (0.2m *v.s.* 10m), spectral (4 *v.s.* 10 bands), and temporal (single-date *v.s.* year-long time series) resolutions. The discrepancy makes the task of integrating this diverse yet complementary information into a unified pixel representation a substantial challenge.

**Domain-Shifts.** The FLAIR dataset spans a large spatio-temporal extent across the entire French metropolitan territory and various seasons over 3 years. This introduces complex *prior-shift* (for instance, there are more vineyards near Bordeaux than in Normandy), and *concept-shift* (variations in roof architecture across regions). This last phenomenon can profoundly impact the appearance of natural and agricultural vegetation, and may also affect the sunlight illumination conditions.

Two camera models were used to capture the aerial images Vexcel's Ultracam Eage Mark3 [32] and IGN's CAMv2 [33]. Although generally minor, this can cause slight differences in resolution and

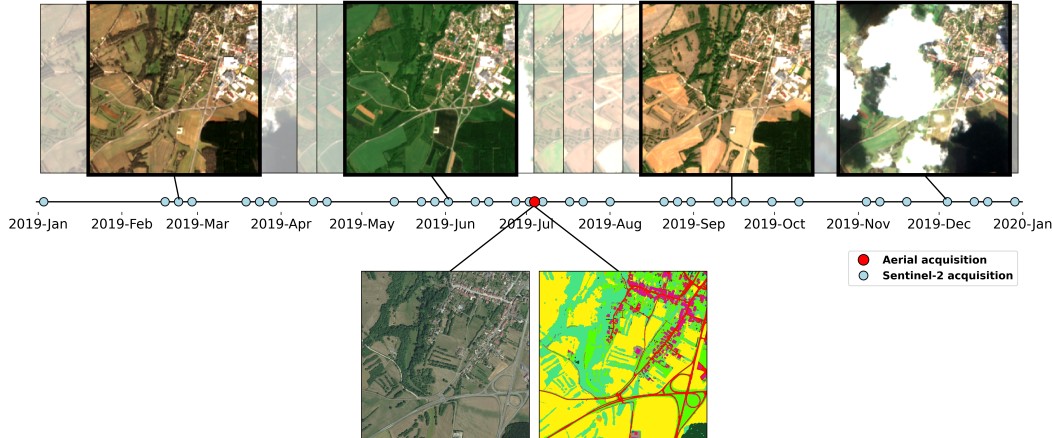

**Figure 4: Satellite Image Time Series.** We represent a year-long satellite time series (*top*), and its associated mono-temporal aerial acquisition and corresponding annotations (*bottom*).

spectral sensitivity. Additionally, all aerial acquisitions undergo radiometric correction to mitigate disparities caused by sunlight and contrast. This can lead to dissimilarities between spatial domains regarding the spectral response of identical materials.

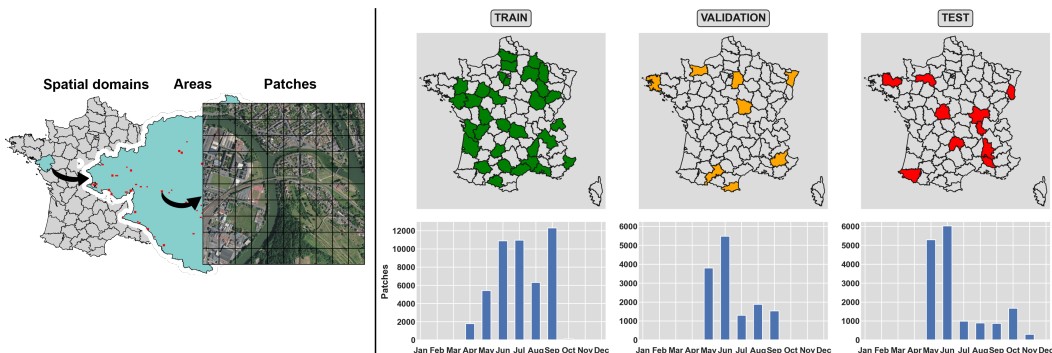

**Figure 5: Spatio-Temporal Distribution.** Spatial units of the FLAIR dataset (*left*), and spatial and temporal distribution of the train / validation / test split (*right*).

## 4  Baselines

We propose a generic yet powerful multi-sensor architecture to serve as a baseline to evaluate the semantic segmentation performance of different approaches.

**General Architecture.** An effective model for our task needs to capture both detailed textures from the aerial images and complex temporal dynamics from the time series. We propose a network architecture named **U-T&T**: U-net with *Textural* and *Temporal* information. As shown in Figure 6, our model consists of two networks: one operating on high-resolution images with four radiometric channels (red, green, blue, infrared) and one elevation channel, and one network operating on time series. Each network follows the state-of-the-art approach for their respective data-source.

- **U-Net (spatial/texture branch)**: We use a U-Net [34] with a ResNet34 backbone model [35] pre-trained on the ImageNet dataset [36]. We add two channels on the first layers to accommodate near-infrared and elevation pixel values. The weights of these two channels are initialized randomly [37]. This U-Net branch comprises approximately 24.4 million parameters.

- **U-TAE (spatio-temporal branch)**: We employ a U-Net with temporal attention (U-TAE) to process the Sentinel-2 imagery [38]. This model is specifically designed to extract multi-scale

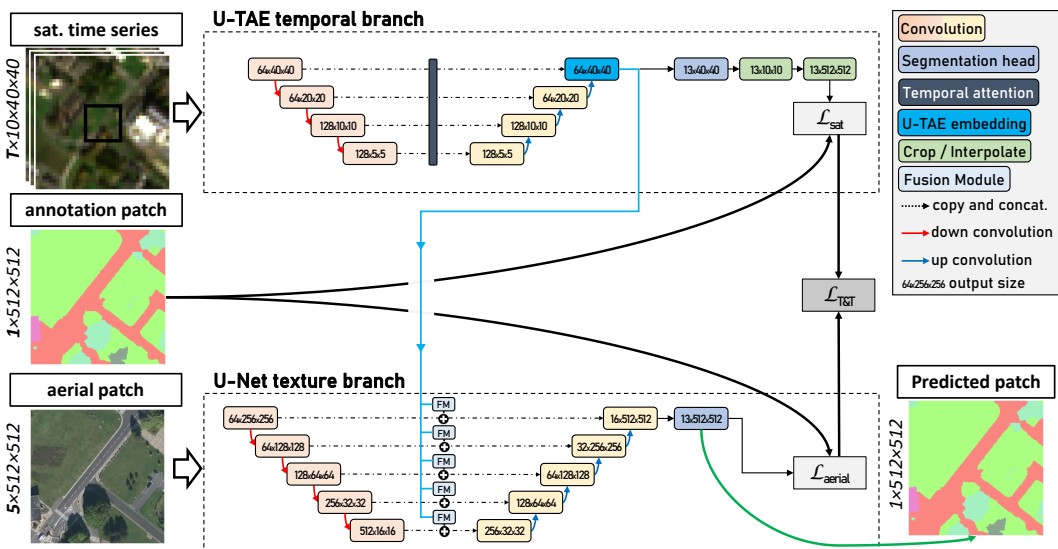

**Figure 6: The U-T&T model.** Our architecture comprises two modules: (i) a U-TAE network extracts spatio-temporal descriptors from the Sentinel-2 time series, and (ii) a U-Net network processes the aerial and elevation images. We merge both feature maps with a branch fusion module. Both branches are supervised simultaneously with dedicated losses $\mathcal{L}_{\text{aerial}}$ and $\mathcal{L}_{\text{sat}}$.

spatio-temporal feature maps from satellite image time series. The U-TAE branch includes approximately 2.9 million parameters.

**Metadata Encoding.**    Metadata can significantly impact the interpretation of remote sensing acquisitions. To allow the network to model this specificity, we compute the following features: (i) spatial coordinates of the center of the patch (with Fourier features), (ii) altitude from sea level, (iii) year of acquisition (one-hot-encoded), and (iv) camera types (one-hot-encoded). These features are then processed with a Multi-Layer Perceptron (MLP) and concatenated channelwise to the coarsest (innermost) feature map of the encoder of the U-Net branch. For more details, refer to the appendix.

**Fusion Module.**    We propose to fuse the multi-scale feature maps from the two branches at different scales. For each level of the U-Net encoder of width $C$ and extent $H \times W$, our proposed fusion module has two sub-parts:

- *Cropped*: This sub-module captures local spatio-temporal information from the image time series. We first crop the pixel-level feature map from the U-TAE to the extent of the aerial image. Then, we apply a spatial convolution with a width of $C$ and interpolate the results to size $H \times W$. Finally, we add the resulting tensor to the intermediate feature map of the U-Net encoder.

- *Collapsed*: This sub-module captures a larger spatial context of the patch from the image time series. We first compute the spatial mean for each channel of the output map of the U-TAE (of size $40 \times 40$). Then, we process the resulting vector with a three-layer MLP to map it to a width of $C$. We add this vector to each pixel of the intermediate feature map of the U-Net encoder.

**Network Supervision.**    The U-Net branch produces a prediction of size $13 \times 512 \times 512$, which can be directly supervised pixelwise with the loss $\mathcal{L}_{\text{aerial}}$, chosen as the categorical cross entropy. The output of the U-TAE branch is of size $13 \times 40 \times 40$, and its extent is larger than the annotation. To adjust this, we first crop it around the aerial image into a tensor of size $13 \times 10 \times 10$, then use bilinear interpolation to map it to the desired $13 \times 512 \times 512$ dimension. This allows us to define its loss $\mathcal{L}_{\text{sat}}$ as the categorical cross entropy as well.

All class weights are set to 1 except for the *other* class, which is set to 0 in both losses. The weight of the class *plowed land* is set to 0 in the loss $\mathcal{L}_{\text{sat}}$ as it corresponds to a transient state of land cover

whose duration is much shorter than the yearly span of the time series. We combine both losses into a single loss $\mathcal{L}_{\text{T\&T}}$ defined as their unweighted sum:

$$\mathcal{L}_{\text{T\&T}} = \mathcal{L}_{\text{aerial}} + \mathcal{L}_{\text{sat}} .$$

**Implementation Details.** The baseline is implemented with PyTorch Lightning [39]. The code for U-Net branch is taken from the segmentation-models-PyTorch library [40], and the U-TAE network is from its official repository [38]. We use the default U-TAE parameters, except for larger widths for the encoder and decoder

The network is optimized with stochastic gradient descent, a batch size of 10, and a learning rate of 0.001. We set the maximum number of epochs to 100 and use early stopping with a patience of 30 epochs. We employ several augmentation strategies, such as cloud removal, temporal averaging, or geometric augmentation; see the appendix for additional details. Our model is trained on a cluster of 12 NVIDIA Tesla V100 GPUs with 32 GB memory.

# 5    Benchmark

**Metric.** We assess the performance of different configurations of our baseline using the mean intersection-over-union (mIoU) on the first 12 classes, excluding the *other* class. We train each model 5 times from scratch, allowing us to compute the standard deviation of the performance.

**Analysis.** We report in Table 3 the performance for several variations of our baseline models. A U-TAE model using only the satellite image time series and whose prediction is upsampled to the resolution of the aerial images leads to much lower performance than its high-resolution counterparts. In contrast, the performance of the multi-sensor U-T&T model is comparable to the simple U-net operating on high-resolution images: 54.7 *v.s.* 54.9. However, when using appropriate augmentation strategies for both, the performance gap widens: 54.9 to 56.9. Specifically, we evaluated the following strategies:

- **FILT.** We remove satellite images with a snow or cloud cover of over 60% according to the meta-data (with a probability threshold of 0.5). This led to a gain of $+0.3$ mIoU point on average.

- **AVG M.** We compute the monthly average of the cloudless satellite acquisitions, ensuring that the time series have at most 12 elements. While the gain is modest ($+0.1$ point), this approach decreased the memory usage and the length discrepancy between locations.

- **MTD.** Our meta-data encoding strategy did not show any impact on either approach.

- **AUG.** We perform geometric augmentations on the aerial images: flip, resize, and random affine transform. This resulted in a $+0.6$ point increase in performance.

Table 4 provides per-class IoU scores for the two best runs of U-Net and U-T&T. We report better results for the U-T&T model for all 12 classes. In particular, we observe significant improvements for classes that particularly benefit from temporal information and broader spatial context, such as bare soil, coniferous, and vineyard classes. Conversely, the improvement is smaller for the water and plowed land classes. These classes can change significantly across the year due to tides, shifting river beds, or harvesting events. Consequently, year-long observation may not bring useful information. See Figure 7 for qualitative illustrations. More examples are provided in the appendices.

**Decoder Architecture.** We evaluated the performance of two other decoder networks in the aerial image branch: FPN [41] and DeepLabV3 [42], while we keep the ResNet34 encoder backbone. We report in Table 5 the results with and without our proposed enhancement strategies. The largest network, DeepLabV3, slightly outperforms the other architecture by 0.4 mIoU points.

**Table 3: Quantitative Evaluation.** Performance of the U-Net and multi-sensor U-T&T architectures on the test set. Results are averages of 5 runs of each configuration. **PARA.**: number of parameters of the network, in million; **EP.**: best validation loss epoch.

| | INPUT | FILT. | AVG M. | MTD | AUG | PARA. | EP. | mIoU |
|---|---|---|---|---|---|---|---|---|
| **UTAE** | sat | ✗ | ✗ | - | - | 2.3 | 16 | **36.1**±0.3 |
| +FILT +AVG M | sat | ✔ | ✔ | - | - | 2.3 | 18 | **36.9**±0.2 |
| **U-Net** | aerial | - | - | ✗ | ✗ | 24.4 | 62 | **54.7**±0.1 |
| *+MTD +AUG* | aerial | - | - | ✔ | ✔ | 24.4 | 52 | **55.2**±0.1 |
| **U-T&T** | aerial+sat | ✗ | ✗ | ✗ | ✗ | 27.3 | 9 | **54.9**±0.7 |
| *+FILT* | aerial+sat | ✔ | ✗ | ✗ | ✗ | 27.3 | 11 | **55.2**±1.4 |
| *+AVG M* | aerial+sat | ✗ | ✔ | ✗ | ✗ | 27.3 | 10 | **55.0**±0.7 |
| *+MTD* | aerial+sat | ✗ | ✗ | ✔ | ✗ | 27.3 | 7 | **54.9**±0.6 |
| *+AUG* | aerial+sat | ✗ | ✗ | ✗ | ✔ | 27.3 | 22 | **55.5**±1.5 |
| *+FILT +AVG M +MTD +AUG* | aerial+sat | ✔ | ✔ | ✔ | ✔ | 27.3 | 21 | **56.9**±1.1 |

**Table 4: Per-Class Evaluation.** We report the classwise IoU for the best run of the U-Net baseline (aerial imagery), and the U-T&T baseline (aerial and satellite imagery).

| model | best | building | perv. | imperv. | bare soil | water | coniferous | deciduous | brushwood | vineyards | herbaceous | agriculture | plowed |
|---|---|---|---|---|---|---|---|---|---|---|---|---|---|
| U-Net | 54.7 | 80.1 | 47.3 | 69.9 | 30.8 | 79.9 | 57.6 | 70.1 | 23.9 | 60.1 | 46.5 | 54.5 | 35.8 |
| U-T&T | **58.6** | **83.5** | **52.0** | **74.0** | **43.7** | **82.2** | **64.2** | **73.7** | **25.6** | **64.6** | **47.17** | **55.2** | **37.0** |

**Table 5: Decoder Architecture.** We report the performance of different network architectures, with and without our proposed enhancement strategies.

| Aerial imagery branch model | enhancements | PARA. | mIoU |
|---|---|---|---|
| U-Net [34] | - | 27.3 | 54.9±0.7 |
| FPN [41] | - | 26.1 | 55.5±1.0 |
| DeepLabV3 [42] | - | 28.8 | 55.8±1.6 |
| U-Net [34] | *+FILT +AVG M +MTD +AUG* | 27.3 | 56.9±1.1 |
| FPN [41] | *+FILT +AVG M +MTD +AUG* | 26.1 | 56.2±0.6 |
| DeepLabV3 [42] | *+FILT +AVG M +MTD +AUG* | 28.8 | 57.3±0.9 |

# 6 Discussion

**Challenges.** The FLAIR dataset was used in two CodaLab [43] scientific challenges, both of which received over 1000 submissions, indicating significant interest. The first challenge, from November 2022 to March 2023, involved domain adaptation while the second challenge, from May to September 2023, incorporated Sentinel-2 time series and introduced a new test dataset. These challenges allowed the scientific community to leverage our extensive labeling effort to evaluate, design, and improve large-scale semantic segmentation methods for multi-sensor Earth observation.

**Limitations.** The FLAIR dataset is limited to metropolitan France. Although France's territory is quite diverse, featuring oceanic, continental, Mediterranean, and mountainous bioclimatic regions, it does not contain tropical or desert area. As a national agency, IGN's focus is limited to France. However, similar efforts by other countries or agencies would increase the diversity of available data and stimulate the design and evaluation of geographically robust methods.

The FLAIR dataset's reliance on purely optical data may limit the applicability of the models trained on it to regions with pervasive cloud cover. Incorporating synthetic aperture radar time series, such as Sentinel-1 time series, may address this limitation and could be considered in a future extension.

**Quality Control.** As the annotations are made through visual interpretation, some errors are unavoidable, especially for classes that are visually hard to distinguish, such as bare soil and pervious surfaces. We manually annotated around 37k randomly selected polygons, hidden from the annotating

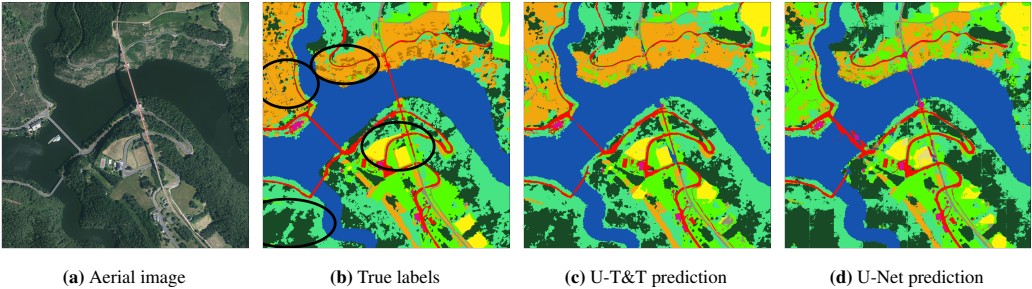

| **(a)** Aerial image | **(b)** True labels | **(c)** U-T&T prediction | **(d)** U-Net prediction |

**Figure 7: Qualitative illustration.** Black ellipses shows areas that are significantly better classified by the multi-sensor approach U-T&T than the standard U-Net model.

teams. This covered a total area of 18.7 km$^2$, equivalent to approximately 0.75% of the entire dataset, or 468 million pixels. If any batch of annotations did not meet a set accuracy criterion of 95%, it was rejected and returned for re-annotation. This iterative process fostered productive exchanges between the annotators and the geography experts from IGN, thereby ensuring a high-quality dataset.

**Ethics.** Releasing a large-scale, high-resolution land-cover dataset openly could raise potential concerns related to privacy, security, and possible misuse. Indeed, detailed information about private properties could be extracted, possibly aiding illegal activities. However, both aerial and satellite images are already publicly available, and we only provide visual interpretations. Furthermore, high-risk areas such as military facilities and nuclear plants are explicitly excluded from the dataset.

FLAIR focuses on the French metropolitan area, which does not well represent countries in arid or tropical climates. This bias could steer scientific efforts towards developing models that perform well in the Western hemisphere while overlooking developing countries. These countries could significantly benefit from automated land-cover tools due to their challenging climates or struggling institutions. IGN is committed to equity and plans future work to focus on more diverse climates, for instance, by planning acquisitions and annotations of French overseas territories.

**Potential Social Impact.** The primary goal of FLAIR is to stimulate the development of robust and scalable tools for automatic land cover. These tools could be instrumental in monitoring and curbing soil artificialization and its catastrophic environmental impacts. Our dataset can also be used to develop and assess the performance of other key geosaptial analysis tasks such as deforestation or sea level monitoring.

By providing a curated and accessible dataset of Earth observations, we also hope to draw the interest of the computer vision community to the challenges of geospatial analysis and contribute to establishing remote sensing data as a standard modality for evaluating machine learning algorithms' performance. The dataset can also facilitate the pre-training of models for other geospatial analysis tasks, such as object detection, super-resolution, or change detection.

## 7 Licence

FLAIR is under the Open Licence 2.0 of Etalab. This licence has been designed to be compatible with any free licence that at least requires an acknowledgement of authorship, and specifically with the previous version of this licence as well as with the following licences: United Kingdom's "Open Government Licence" (OGL), Creative Commons' "Creative Commons Attribution" (CC-BY) and Open Knowledge Foundation's "Open Data Commons Attribution" (ODC-BY).

## 8 Acknowledgment

The experiments conducted in this study were performed using HPC/AI resources provided by GENCI-IDRIS (Grant 2022-A0131013803). This work was supported by the European Union through the project "Copernicus / FPCUP" as well as by the French Space Agency (CNES) and Connect by CNES. The authors would like to acknowledge the valuable support and resources provided by these organizations.

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

# A    Appendix

## A.1    Benefits of the multi-source approach

Sentinel-2 imagery are synergistic approach with VHR aerial images for land cover mapping, as each source has a unique advantage allowing to distinguishing nuanced semantic classes, a critical need in detailed geospatial analysis. Some of the main benefits of integrating Sentinel-2 are:

- **Increased spectral resolution:** Unlike aerial acquisitions that generally contain only four spectral bands (with a single one in the infrared), Sentinel-2 is furnished with a 10-band multispectral imager. This includes bands in the near-infrared spectrum, which prove essential for discerning vegetation phenology [44].

- **Multi-temporal resolution:** Sentinel-2 provides a consistent yearly time series. This capability allows our model to trace the temporal progression of each pixel's spectral response, proving invaluable in distinguishing between similar plant species, as depicted in Figure 2. As an illustrative example, while an "agricultural land" and a "herbaceous surface" might appear identical during specific times (exhibiting low herbaceous vegetation), the agricultural land remains barren of vegetation during other parts of the year. VHR aerial acquisitions, in contrast, are limited to single-date images.

- **Larger spatial context:** The coarser spatial resolution of Sentinel-2 (10 m) compared to aerial images (20 cm) provides an unexpected advantage. By offering a broader context, Sentinel-2 enables our model to harness wider receptive fields. Consequently, each 102x102m aerial patch is linked with a Sentinel-2 image time series spanning a 400x400m area.

- **Spectral Consistency:** The Sentinel-2 time series benefits from consistent spectral calibration, which aids in countering the radiometric inconsistencies introduced during the BD Ortho's correction process.

## A.2    Sentinel-2 Time Series

Table 6 indicates the original bands acquired by the Sentinel-2 satellites and considered in the FLAIR dataset. The images were downloaded from the Sinergise API [45] as Level-2A products (Bottom-Of-the-Atmosphere reflectances) which are atmospherically corrected using the Sen2Cor algorithm [31].
[1] Sentinel-2 sensor acquires images at 10, 20 and 60 m spatial resolutions. The 60 m bands mainly intended for atmospheric corrections are not taken into account and the 20 m bands are resampled during data retrieval to 10 m by the nearest interpolation method.

**Table 6: Sentinel-2 spatial and spectral resolutions.** Original spatial and spectral resolutions of Sentinel-2 images along with the correspondence between original band number and the distributed data.

| Original Band number | FLAIR band number | Central wavelength (nm) | Bandwidth (nm) | Original Spatial resolution (m) | FLAIR Spatial resolution (m) |
|---|---|---|---|---|---|
| 2 | 1 | 490 | 65 | 10 | 10 |
| 3 | 2 | 560 | 35 | 10 | 10 |
| 4 | 3 | 665 | 30 | 10 | 10 |
| 5 | 4 | 705 | 15 | 20 | 10 |
| 6 | 5 | 740 | 15 | 20 | 10 |
| 7 | 6 | 783 | 20 | 20 | 10 |
| 8 | 7 | 842 | 115 | 10 | 10 |
| 8a | 8 | 865 | 20 | 20 | 10 |
| 11 | 9 | 1610 | 90 | 20 | 10 |
| 12 | 10 | 2190 | 180 | 20 | 10 |

Table 7 indicates the cloud & snow probability masks provided as separate files alongside the Sentinel-2 acquisitions. It should be noted that cloud detection in satellite images is a complex task because

---

[1]More advanced algorithms [46] could be beneficial.

of the diversity of clouds (thin, scattered clouds). As a result, probability masks can contain errors, notably confusion with surfaces with a high albedo and close to the top of a cloud, as is the case with the roofs of industrial buildings.

**Table 7: Provided cloud and snow masks.**

| Mask | FLAIR band number | Original Spatial resolution (m) | FLAIR Spatial resolution (m) |
|---|---|---|---|
| Snow probability (SNW) | 1 | 20 | 10 |
| Cloud probability (CLD) | 2 | 20 | 10 |

Table 8 provides information about the number of dates included in the filtered Sentinel-2 time series for the train and test datasets. On average, each area is acquired on 55 dates over the course of a year by satellite imagery.

**Table 8: Sentinel-2 Time series length.** Number of acquisitions (dates) in the Sentinel-2 times series of one year (corresponding to the year of aerial imagery acquisition).

| | acquisitions per super-area | | |
|---|---|---|---|
| Sentinel-2 time series (1 year) | min | max | mean |
| train dataset | 20 | 100 | 55 |
| test dataset | 20 | 114 | 55 |

Note that cloudy dates are not suppressed from the time series. Instead, the masks are provided and can be used to filter the cloudy dates if needed.

The spatial size of Sentinel-2 time series has been empirically determined and set to 40. Nevertheless, we provided in this dataset wider areas than the $40 \times 40$ used for our baseline. However, there is a limit of 110 pixels for edge patches. The choice of time series spatial size has an impact on the spatial context provided to both the U-TAE and U-Net branches through the *collapsed* fusion sub-module [47].

### A.3   Semantic classes

Overall semantic class number of pixels and frequency of the FLAIR dataset are provided in Table 9. The class distribution in percentages of the train and test sets are presented in Figure 8. The detailed description of the original semantic classes is provided in Table 10.

The ground truth labels are based on photo-interpretation of the aerial imagery at 20 cm and has been manually produced by experts following a call for tenders from the IGN. An initial spatial multi-level image segmentation approach using PYRAM [48] was applied, simplifying the labeling at the small cluster level. This segmentation was modified interactively when deemed appropriate.

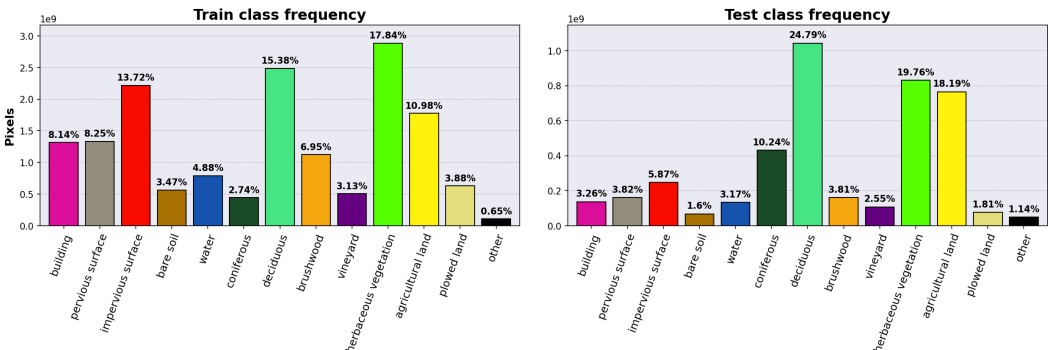

**Figure 8: Class distribution** of the train dataset (*left*) and test dataset (*right*).

**Table 9: Details about the semantic classes of the main nomenclature of the FLAIR dataset** and their corresponding label values, frequency in pixels and percentage among the entire dataset.

| | Class | Label Value | Pixels | % |
|---|---|---|---|---|
| | building | 1 | 1,453,245,093 | 7.13 |
| | pervious surface | 2 | 1,495,168,513 | 7.33 |
| | impervious surface | 3 | 2,467,133,374 | 12.1 |
| | bare soil | 4 | 629,187,886 | 3.09 |
| | water | 5 | 922,004,548 | 4.52 |
| | coniferous | 6 | 873,397,479 | 4.28 |
| | deciduous | 7 | 3,531,567,944 | 17.32 |
| | brushwood | 8 | 1,284,640,813 | 6.3 |
| | vineyard | 9 | 612,965,642 | 3.01 |
| | herbaceous vegetation | 10 | 3,717,682,095 | 18.24 |
| | agricultural land | 11 | 2,541,274,397 | 12.47 |
| | plowed land | 12 | 703,518,642 | 3.45 |
| | other | **>13** | 153,055,302 | 0.75 |

**Table 10: Semantic classes of the FLAIR dataset.**

## Class description

*Note: as previously stated, semantic classes are assigned on the cluster level. In a given aerial image, only observable objects are labeled, whereby temporal aspects are not taken into consideration.*

**Anthropized surfaces without vegetation (1, 2, 3, 13 and 18)**

*Class 1 – building* includes not only buildings but also other types of constructions such as towers, agricultural silos, water towers and dams. Greenhouses (class 18) are an exception.
*Class 2 – pervious surface* defined as man-made bare soils covered with mineral materials (*e.g.* gravel, loose stones) and considered to be pervious. It includes pervious transport networks (*e.g.* gravel pathways, railways), quarries, landfills, building sites and coastal ripraps.
*Class 3 – impervious surface* is defined as man-made bare soils that are impervious due to their building materials (e.g. concrete, asphalt, cobblestones). It includes roadways, parking lots, and certain types of sports fields.
*Class 13 – swimming pool* is defined as man-made artificial (open-air) swimming pools. It is not included in class 5 (water).
*Class 18 – greenhouse* although it can be considered as a building, is given a distinct label. Greenhouses are a class of their own and are not part of class 1.

**Natural areas without vegetation (4, 5 and 14)**

*Class 4 – bare soil* defined as natural permanently bare soils. These natural soils remain without vegetation throughout the year and generally are covered with sand, pebbles, rocks or stones. Examples of natural bare soils are frequently found in coastal, mountainous and forested areas.
*Class 5 – water* is defined as areas covered by water, such as sea, rivers, lakes and ponds. An exception are swimming pools (class 13).
*Class 14 – snow* refers to surfaces covered by snow. It is an extremely rare class as the images are taken in the summertime and only very few regions in France are covered with snow year-round.

**Woody natural vegetation surfaces (6, 7, 8, 15, 16 and 17)**

*Class 6 – coniferous*, is defined as trees identifiable as coniferous (pines, firs, cedars, cypress trees, ...) and taller than 5 m.
*Class 7 – deciduous* is defined as trees identifiable as deciduous (oaks, beeches, birches, chestnuts, poplars, ...) and taller than 5 m.
*Class 8 – brushwood* refers to natural woody surfaces with a vegetation less than 5 m high. It includes short and young trees, brushwood, shrublands, mountain moors and abandoned agricultural lands.
*Class 15 – clear-cut*, is defined as forest areas, in which the trees have been cut down and harvested.
*Class 16 – ligneous* is an extremely rare class used to describe forest areas with a homogeneous representation of either coniferous or deciduous trees.
*Class 17 – mixed* is an extremely rare class used to describe forest areas with heterogeneous trees for which the types of trees (coniferous/ deciduous) cannot be determined with sufficient certainty.

**Agricultural surfaces (9, 11 and 12)**

*Class 9 – vineyard* despite being an agricultural use of the land, are assigned a class apart, a reason being their rather distinctive land cover characteristics.
*Class 11 – agricultural land* encompasses various different agricultural classes. For example, besides major crops, it also includes permanent and temporary grasslands with agricultural use. Vineyards (class 9) are not included in this class.
*Class 12 – plowed land* is defined as agricultural land with no visible vegetation (*e.g.* recently plowed and freshly harvested land).

**Herbaceous surfaces (10)**

*Class 10 – herbaceous vegetation* defines herbaceous surfaces that are not intensively exploited for agriculture purposes. This class includes ornamental lawns (e.g. gardens, public parks), recreational fields (*e.g.* used for sport), natural herbaceous areas in forested or mountainous areas, non-cultivated grass in agricultural areas or along transportation networks.

## A.4 Aerial imagery and spatial domains

Within a spatial domain, all aerial acquisitions are radiometrically corrected to reduce disparities in sunlight and contrast. Nonetheless, this homogenization is not applied equally across all the different spatial domains as can be seen in Figure 9. As opposed to satellite imagery, the pixel intensity in the image channels can therefore not be considered as a physical measure.

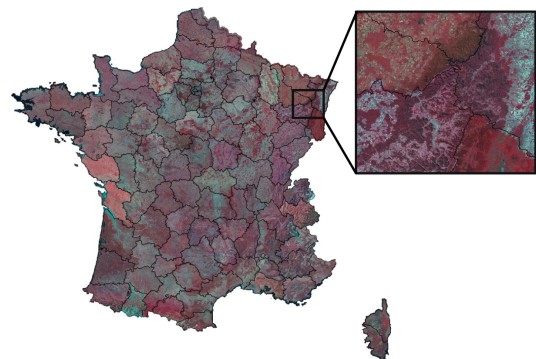

**Figure 9: Radiometric discrepancies of the aerial images between domains.** The 3 channels image displayed is a composite of Near-Infrared, Red and Green spectral information.

## A.5 Benchmark architecture

### A.5.1 U-Net (spatial/texture branch)

We choose a U-Net architecture [34] with a ResNet34 encoder backbone (pre-trained on the ImageNet dataset [36]) for a total of $\approx 24.4\,$M parameters and rely on the implementation available in the *segmentation-models-pytorch* library [40] and trained with the PyTorch lightning [39] framework. The architecture employed is illustrated in Figure 10.

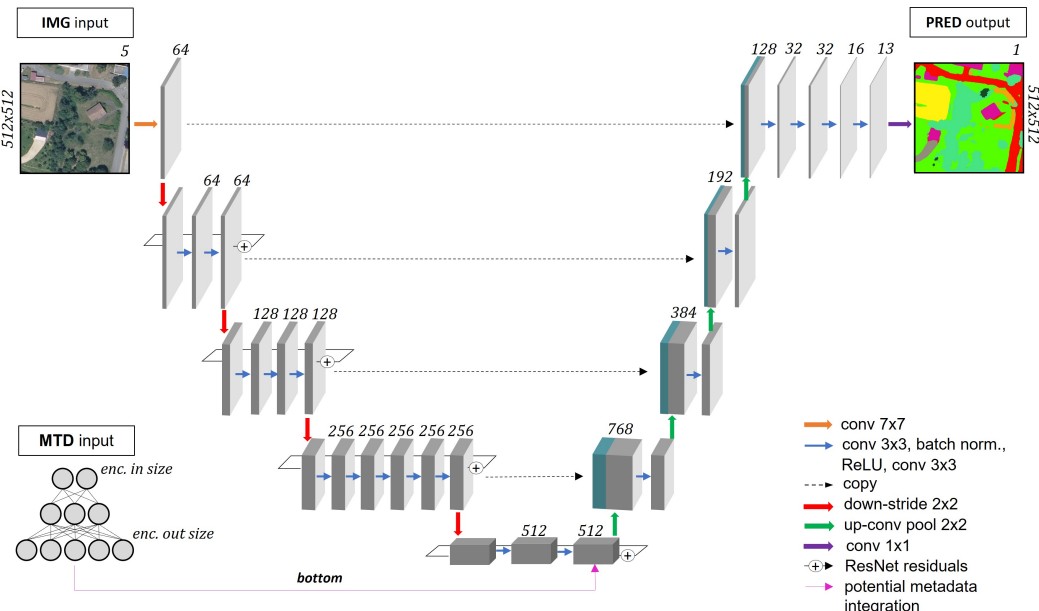

**Figure 10: U-Net architecture used for the baseline**. **IMG** = input image; **MTD** = input metadata; **PRED** = prediction output. One potential and traditional approach to integrate the metadata would be to add a Multi-layer Perceptron for encoding and add the output to the output of the last layer of the encoder or as an additional band to the IMG input.

Concerning the exploitation of metadata, a simple approach has been tested [49]. The strategies explored have a first step of metadata encoding: positional encoding of spatial and temporal information and one-hot-encoding for camera type and aerial image acquisition year. A shallow MLP with dropout (probability of 0.4) and ReLU activation is then defined to jointly encode the metadata and to provide a specified output size. Subsequently, multiple different integration strategies with the current ResNet34/U-Net segmentation architecture are possible. We have chosen a commonly employed strategy (depicted as 'bottom' in Figure 10) consists in matching the MLP output size to the output size of the last layer of the ResNet34 encoder. The two vectors (encoded metadata and encoded images) can then be added and fed into the first layer of the architecture's decoder. Strategies following similar approaches that add the MLP encoded output at different positions in the architecture's encoder or decoder parts (*e.g.*, after the first input convolution layer, with the last decoder layer, or even added as a sixth channel to the input image) are possible. A positional encoding of size 32 is used specifically for encoding the geographical location information.

The exploitation of metadata deserves to be studied more by the computer vision community, as it could bring real gains by taking advantage of the specificity of remote sensing data.

### A.5.2 Fusion module of the U-T&T model

A Fusion Module is employed within the U-T&T baseline model to integrate the feature maps from satellite time-series (with broader spatial extent) into the feature maps from the aerial imagery branch. The details of this module can be seen in Figure 11. Within the *Fusion Module*, two sub-modules (*cropped* and *collapsed*) have different purposes and focus on distinct aspects: the spatio-temporal information and the spatial context. This *Fusion Module* is applied to match with each feature maps of the U-Net encoder.

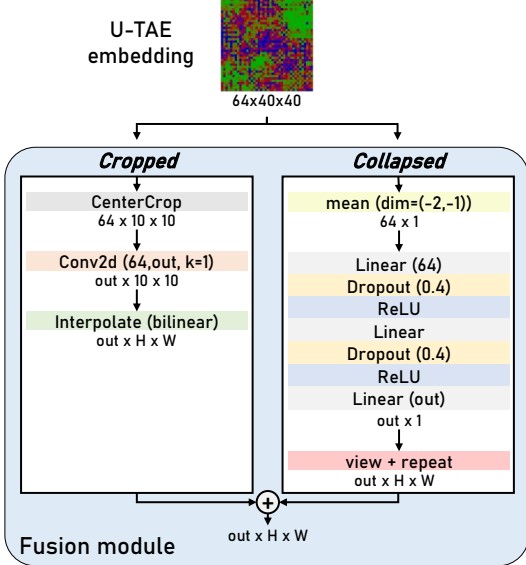

**Figure 11: Fusion module.** This module takes as input the last U-TAE embeddings. It is applied to each stage of the U-Net encoder feature maps. *out* corresponds to the channel size of the U-Net encoder feature map and *H* and *W* to the corresponding spatial dimensions.

### A.5.3 Data augmentation

By introducing variance in the dataset, image data augmentation helps to prevent overfitting and provides trained models with enhanced generalization capabilities.

For our baseline, only geometric transformations are explored using the *Albumentation* library. Vertical and horizontal flips, and random rotations of 0, 90, 180 or 270 degrees are tested. A data augmentation probability of 0.5 is used.

## A.6 Benchmark results

### A.6.1 Official data split of the FLAIR dataset

The following per domain split of the data has been used for the experiments:

| | |
|---|---|
| TRAIN: | D006, D007, D008, D009, D013, D016, D017, D021, D023, D030, D032, D033, D034, D035, D038, D041, D044, D046, D049, D051, D052, D055, D060, D063, D070, D072, D074, D078, D080, D081, D086, D091 |
| VALIDATION: | D004, D014, D029, D031, D058, D066, D067, D077 |
| TEST: | D015, D022, D026, D036, D061, D064, D068, D069, D071, D084 |

### A.6.2 Extra results

We present in Table 11 the performance of a U-TAE network using only satellite image time series upsampled spatially to the resolution of the aerial images. The lower resolution of this sensor leads to signficantly worse results.

**Table 11: Quantitative Evaluation.** Performance of the U-Net and multi-sensor U-T&T architectures on the test set. Results are averages of 5 runs of each configuration. **PARA.**: number of parameters of the network, in million; **EP.**: best validation loss epoch.

| | INPUT | FILT. | AVG M. | MTD | AUG | PARA. | EP. | mIoU |
|---|---|---|---|---|---|---|---|---|
| **U-Net** | aerial | - | - | ✗ | ✗ | 24.4 | 62 | **54.7**±0.1 |
| *+MTD* | aerial | - | - | ✔ | ✗ | 24.4 | 59 | **54.7**±0.2 |
| *+MTD +AUG* | aerial | - | - | ✔ | ✔ | 24.4 | 52 | **55.2**±0.1 |
| **U-TAE** | sat | ✗ | ✗ | - | - | 2.3 | 16 | **36.1**±0.3 |
| *+FILT* | sat | ✔ | ✗ | - | - | 2.3 | 14 | **35.9**±0.9 |
| *+FILT +AVG M* | sat | ✔ | ✔ | - | - | 2.3 | 18 | **36.9**±0.2 |
| **U-T&T** | aerial+sat | ✗ | ✗ | ✗ | ✗ | 27.3 | 9 | **54.9**±0.7 |
| *+FILT* | aerial+sat | ✔ | ✗ | ✗ | ✗ | 27.3 | 11 | **55.2**±1.4 |
| *+AVG M* | aerial+sat | ✗ | ✔ | ✗ | ✗ | 27.3 | 10 | **55.0**±0.7 |
| *+MTD* | aerial+sat | ✗ | ✗ | ✔ | ✗ | 27.3 | 7 | **54.9**±0.6 |
| *+AUG* | aerial+sat | ✗ | ✗ | ✗ | ✔ | 27.3 | 22 | **55.5**±1.5 |
| *+FILT +AVG M +MTD +AUG* | aerial+sat | ✔ | ✔ | ✔ | ✔ | 27.3 | 21 | **56.9**±1.1 |

Figure 12 illustrates the confusion matrix of the best U-T&T model. This confusion matrix is derived by combining all individual confusion matrices per patch and is normalized by rows. The analysis of the confusion matrix shows that the best U-T&T model achieves accurate predictions with minimal confusion in the majority of classes. However, when it comes to natural areas such as *bare soil* and *brushwood*, although there is improvement due to the use of Sentinel-2 time series data, a certain level of uncertainty remains. These classes exhibit some confusion with semantically similar classes, indicating the challenge of accurately distinguishing them.

More qualitative examples can be found in Figure 13.

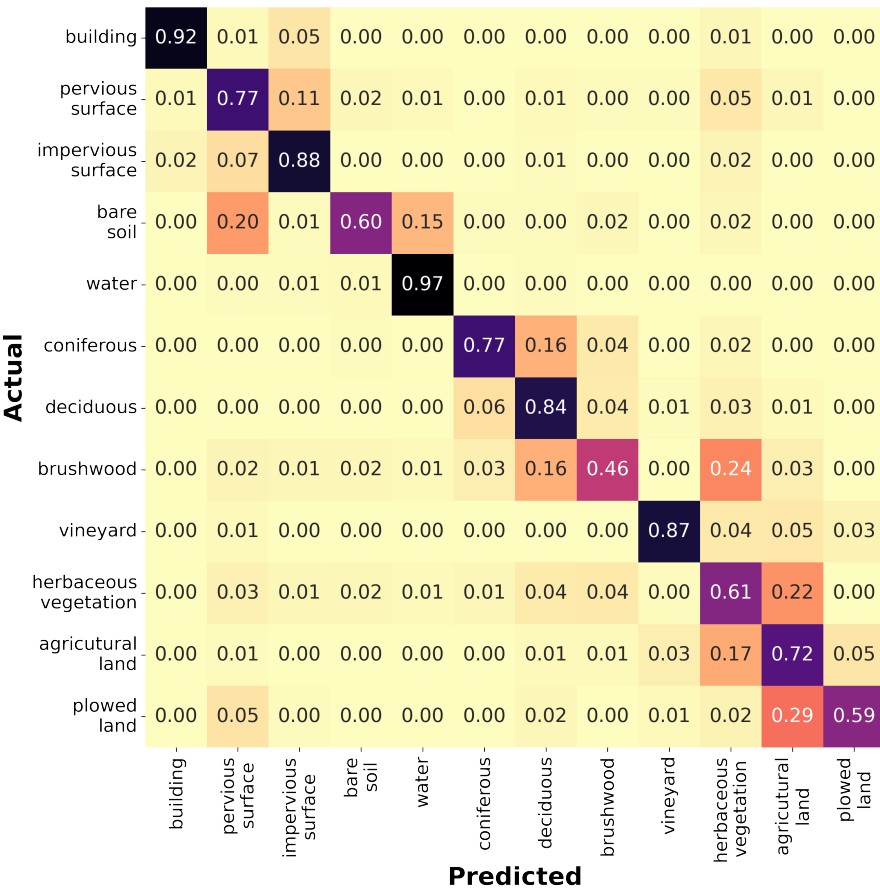

**Figure 12: U-T&T best model confusion matrix of the test dataset**. The matrix is normalized by rows.

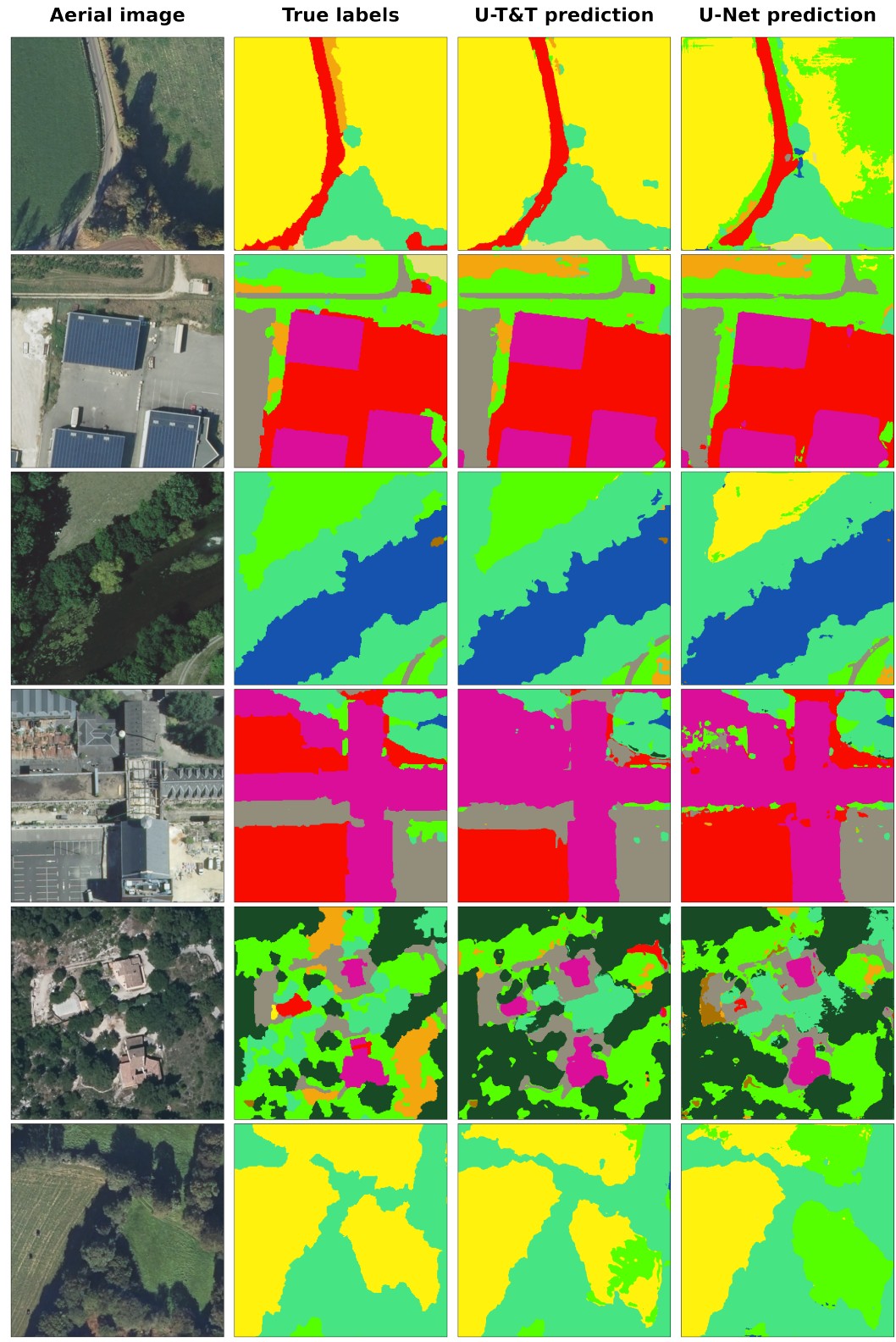

**Figure 13: Illustration of patch-wise results.** Random results on the FLAIR dataset for the multi-sensor approach U-T&T than the standard U-Net model.

# B Datasheet for FLAIR dataset

## B.1 Motivation

Q1 **For what purpose was the dataset created?** Was there a specific task in mind? Was there a particular gap that needed to be filled? Please provide a description.

- The FLAIR dataset is created to train and evaluate models that can predict very-high-resolution land cover maps from diverse data sources with heterogeneous spatial, temporal, and spectral resolutions. The main gap we are addressing is the lack of large-scale data with high-definition annotations.

Q2 **Who created the dataset (e.g., which team, research group) and on behalf of which entity (e.g., company, institution, organization)?**

- This dataset is presented by the French National Institute of Geographical and Forest Information (IGN), a French public state administrative establishment aiming to produce and maintain geographical information for France. The IGN has the mission to document and measure land-cover on French territory and provides referential geographical datasets, including very-high-resolution aerial images and topographic maps. IGN produces reference data and carries out innovation, research and teaching activities. As part of its innovation activities, the IGN provides the FLAIR dataset to democratize access to large-scale open powerful machine learning models through the research and development of open-source resources.

Q3 **Who funded the creation of the dataset?** If there is an associated grant, please provide the name of the grantor and the grant name and number.

- The funding of the FLAIR dataset is 100% public. This work was sponsored by the Ministry of Ecological Transition (more specifically the Directorate for Planning, Housing and Nature *Direction générale de l'aménagement, du logement et de la nature*) and the Fund for the transformation of public action (*Fonds pour la transformation de l'action publique*) from the Minister of the Civil Service. The IGN is funded by the French Ministry of Ecological Transition and the French Ministry of Agriculture.

Q4 **Any other comments?**

- [N/A]

### B.2 Composition

Q5 **What do the instances that comprise the dataset represent (e.g., documents, photos, people, countries)?**

- We provide aerial image with corresponding land cover segmentation along with Sentinel-2 satellite image time series around each aerial patch. The acquisitions are taken from 916 unique areas distributed across 50 French spatial domains (**départements**), covering approximately 817 $km^2$. The test labels will be released at the end of the second challenge hosted on CodaLab. We made our baseline codes openly available on the FLAIR GitHub page (https://github.com/IGNF/FLAIR-2-AI-Challenge).

Q6 **How many instances are there in total (of each type, if appropriate)?**

- We provide 77,762 triplet aerial image, Sentinel-2 time and land cover segmentation. The FLAIR dataset encompasses 20,384,841,728 annotated pixels at a spatial resolution of 0.20 m from aerial imagery with a 19 classes land cover. For each area, a comprehensive one-year record of Sentinel-2 acquisitions is also provided. A further overview of the statistics may be seen in the following annexes.

Q7 **Does the dataset contain all possible instances or is it a sample (not necessarily random) of instances from a larger set?**

- The FLAIR dataset, derived from a larger dataset obtained by IGN for cartographic production upon the request of the French government, serves as a representative sample encompassing approximately one-third of the available data. While the complete dataset covers 64 spatial domains, the FLAIR dataset focuses on 50 domains by excluding

contiguous domains and intra-domain areas. Nevertheless, the selected 50 domains offer comprehensive representation in terms of land cover classes, acquisition dates, and macro-climates, and encompass the metadata associated with the entire dataset. The expertise of IGN was leveraged to ensure the selection of a dataset that is representative and informative.

Q8 **What data does each instance consist of?**

- Each instance consists of an aerial image. Each image is $512 \times 512$ in size with a resolution of 20 cm per pixel, and feature $4$ spectral channels: red, blue, green, and near-infrared along with an elevation value as fifth channel. Each patch is associated with a satellite image time series from the Sentinel-2 constellation (Drusch et al., 2012) of size $40 \times 40$ with a $10$ m pixel resolution, centered around the aerial image. Each pixel from the Sentinel-2 sequences is characterized by $10$ spectral bands.

Q9 **Is there a label or target associated with each instance?**

- [Yes] We provide a complete pixel-precise land cover segmentation per image (19 classes).

Q10 **Is any information missing from individual instances?**

- [No]

Q11 **Are relationships between individual instances made explicit (e.g., users' movie ratings, social network links)?**

- [No]

Q12 **Are there recommended data splits (e.g., training, development/validation, testing)?**

- Yes, we provide data splits for reproducing the results of the baselines. The test split has been explicitly selected to address the complex domain shifts of geospatial data.

Q13 **Are there any errors, sources of noise, or redundancies in the dataset?**

- As the annotations are made through visual interpretation with quality control, some errors are unavoidable, especially for classes that are visually hard to distinguish. Internal quality control with multiple annotations has been performed to limit such errors. There are no redundancies in the dataset, each image covers a distinct area.

Q14 **Is the dataset self-contained, or does it link to or otherwise rely on external resources (e.g., websites, tweets, other datasets)?**

- This dataset is self-contained and will be stored and distributed by the IGN, a public institute. The dataset is under the Open Licence 2.0 of Etalab.

Q15 **Does the dataset contain data that might be considered confidential (e.g., data that is protected by legal privilege or by doctor–patient confidentiality, data that includes the content of individuals' non-public communications)?**

- [No] . The building class does not contain information that would not be available in other open-access sources, such as the cadaster. We have specifically avoided high-risk areas such as military installations or nuclear plants.

Q16 **Does the dataset contain data that, if viewed directly, might be offensive, insulting, threatening, or might otherwise cause anxiety?** *If so, please describe why.*

- [No]

Q17 **Does the dataset relate to people?**

- The dataset may feature pedestrian or individuals, but the resolution of 20cm/pixel and the aerial perspective is not sufficient to recognize them uniquely.

Q18 **Does the dataset identify any subpopulations (e.g., by age, gender)?**

- [No]

Q19 **Is it possible to identify individuals (i.e., one or more natural persons), either directly or indirectly (i.e., in combination with other data) from the dataset?**

- [No] . The resolution of 20cm/pixel and the aerial perspective is insufficient to recognize them uniquely.

Q20 **Does the dataset contain data that might be considered sensitive in any way (e.g., data that reveals racial or ethnic origins, sexual orientations, religious beliefs, political opinions or union memberships, or locations; financial or health data; biometric or genetic data; forms of government identification, such as social security numbers; criminal history)?**

- [No]

Q21 **Any other comments?**

- [No]

### B.3 Collection Process

Q22 **How was the data associated with each instance acquired?**

- The aerial images are sampled from the ORTHO HR® imagery collection. It is a mosaic of all the individual images taken during an aerial survey done by IGN and mapped onto a cartographic coordinate reference system. The individual images are projected to the RGE ALTI® DTM, which provides solely the altitude of the ground.
- The Sentinel-2 time series were downloaded from the Sinergise Sentinel-Hub API as Level-2A products (see annexes for more information).

Q23 **What mechanisms or procedures were used to collect the data (e.g., hardware apparatus or sensor, manual human curation, software program, software API)?**

- The IGN selected several acquisition companies through a call for tender with strict specifications.

Q24 **If the dataset is a sample from a larger set, what was the sampling strategy (e.g., deterministic, probabilistic with specific sampling probabilities)?**

- The sampling strategy involved class frequency, acquisition dates distribution, radiometric histogram analysis and geographical location spread. The final sampling based on these comprehensive variables was made manually by experts at the IGN.

Q25 **Who was involved in the data collection process (e.g., students, crowdworkers, contractors) and how were they compensated (e.g., how much were crowdworkers paid)?**

- IGN contracted geography experts from the private sector selected through a public call for tender to annotate the dataset. The quality control of the dataset was carried out by geography experts affiliated with IGN. The creation of the dataset was facilitated by researchers and developers employed by IGN under their work contracts.

Q26 **Over what timeframe was the data collected? Does this timeframe match the creation timeframe of the data associated with the instances (e.g., recent crawl of old news articles)?**

- The collection of aerial imagery spanned from 2018 to 2021, which coincides with the duration required for an aerial survey to encompass the entirety of the French territory. Annotations were then applied to the aerial images, aligning with the same time frame. Subsequently, the dataset was created in 2022 after the final processing for both the aerial imagery and annotations.

Q27 **Were any ethical review processes conducted (e.g., by an institutional review board)?**

- [No]

Q28 **Does the dataset relate to people?**

- [No]

Q29 **Did you collect the data from the individuals in question directly, or obtain it via third parties or other sources (e.g., websites)?**

- [N/A]

Q30 **Were the individuals in question notified about the data collection?**

- [N/A]

**Q31 Did the individuals in question consent to the collection and use of their data?**

- [N/A]

**Q32 If consent was obtained, were the consenting individuals provided with a mechanism to revoke their consent in the future or for certain uses?**

- [N/A]

**Q33 Has an analysis of the potential impact of the dataset and its use on data subjects (e.g., a data protection impact analysis) been conducted?**

- [No]

**Q34 Any other comments?**

- [No]

### B.4 Preprocessing, Cleaning, and/or Labeling

**Q35 Was any preprocessing/cleaning/labeling of the data done (e.g., discretization or bucketing, tokenization, part-of-speech tagging, SIFT feature extraction, removal of instances, processing of missing values)?**

- [No]

**Q36 Was the "raw" data saved in addition to the preprocessed/cleaned/labeled data (e.g., to support unanticipated future uses)?** *If so, please provide a link or other access point to the "raw" data.*

- [No]

**Q37 Is the software used to preprocess/clean/label the instances available?**

- [No]

**Q38 Any other comments?**

- [No]

### B.5 Uses

**Q39 Has the dataset been used for any tasks already?**

- The optical images of FLAIR train split were used for two data challenges ran in 2022 and 2023 by IGN.
- Marsocci et al., 2023 used a subset of FLAIR for to evaluate techniques for unsupervised domain adaptation.

**Q40 Is there a repository that links to any or all papers or systems that use the dataset?**

- [Yes] . We propose below a list of scientific publications and systems that use FLAIR dataset:
  - Garioud et al., 2022 provides a technical description of the FLAIR aerial imagery dataset;
  - Garioud et al., 2023 provides insight on the multi-sensor fusion of aerial and satellite imagery;
  - Marsocci et al., 2023 experiments remote sensing unsupervised domain adaptation using geographical coordinates on a subset of the FLAIR dataset.

**Q41 What (other) tasks could the dataset be used for?**

- We encourage future researchers to use FLAIR dataset for several tasks. Particularly, we see applications in land cover segmentation and multi-sensor fusion. Due to the breadth of the data, it also offers a unique opportunity for pre-training of models for other geospatial analysis tasks with low resource, such as object detection, super-resolution, or change detection.

**Q42 Is there anything about the composition of the dataset or the way it was collected and preprocessed/cleaned/labeled that might impact future uses?**

- This dataset is geographically limited to metropolitan France. Although France's territory is quite diverse, featuring oceanic, continental, Mediterranean, and mountainous bioclimatic regions, it does not contain tropical or desert areas.
- The FLAIR dataset's reliance on purely optical data may limit the applicability of the models trained on it to regions with pervasive cloud cover.

**Q43 Are there tasks for which the dataset should not be used?**

- [No] .

**Q44 Any other comments?**

- [No] .

### B.6 Distribution

**Q45 Will the dataset be distributed to third parties outside of the entity (e.g., company, institution, organization) on behalf of which the dataset was created?**

- [Yes] the dataset will be open-source.

**Q46 How will the dataset be distributed (e.g., tarball on website, API, GitHub)?**

- The data will be available through *.zip* files available on the FLAIR project page hosted on GitHub (`https://ignf.github.io/FLAIR/`).

**Q47 When will the dataset be distributed?**

- All data with the exception of the test split is presently accessible by registering for an ongoing challenge hosted on Codalab. The entire dataset, including the test split, will be released under an open-source license on the FLAIR project page in early October 2023.

**Q48 Will the dataset be distributed under a copyright or other intellectual property (IP) license, and/or under applicable terms of use (ToU)?** *If so, please describe this license and/or ToU, and provide a link or other access point to, or otherwise reproduce, any relevant licensing terms or ToU, as well as any fees associated with these restrictions.*

- [Yes] . The data is governed by the Open Licence 2.0 of Etalab (`https://www.etalab.gouv.fr/wp-content/uploads/2018/11/open-licence.pdf`).

**Q49 Have any third parties imposed IP-based or other restrictions on the data associated with the instances?**

- [No]

**Q50 Do any export controls or other regulatory restrictions apply to the dataset or to individual instances?**

- [No]

**Q51 Any other comments?**

- [No]

### B.7 Maintenance

**Q52 Who will be supporting/hosting/maintaining the dataset?**

- IGN will support hosting of the dataset and metadata.

**Q53 How can the owner/curator/manager of the dataset be contacted (e.g., email address)?**

- `ai-challenge@ign.fr`

**Q54 Is there an erratum?**

- [No] . There is no erratum for our initial release. Errata will be documented as future releases on the dataset website.

**Q55 Will the dataset be updated (e.g., to correct labeling errors, add new instances, delete instances)?**

- Additional modalities (*e.g.*, supplementary satellite, aerial, UAV-based imagery) may be added to the FLAIR dataset.

Q56 **If the dataset relates to people, are there applicable limits on the retention of the data associated with the instances (e.g., were individuals in question told that their data would be retained for a fixed period of time and then deleted)?**

- N/A

Q57 **Will older versions of the dataset continue to be supported/hosted/maintained?**

- [Yes] . We are dedicated to providing ongoing support for the FLAIR dataset.

Q58 **If others want to extend/augment/build on/contribute to the dataset, is there a mechanism for them to do so?**

- Proposed extensions or corrections to the FLAIR dataset may be submitted to the providers for consideration. The IGN will assess the feasibility of incorporating the suggested modifications, considering factors such as data licensing, maintenance requirements, and relevance.

Q59 **Any other comments?**

- [No] .

