# OpenReview forum: "FLAIR : a Country-Scale Land Cover Semantic Segmentation Dataset From Multi-Source Optical Imagery"
_NeurIPS.cc/2023/Track/Datasets_and_Benchmarks — NeurIPS 2023 Datasets and Benchmarks Poster_

### Official Review · Reviewer_j7rF · 2023-07-21
**The comments and suggestions on FLAIR dataset paper.**

**Rating:** 5
**Confidence:** 4
**Clarity:** Yes.

**Strengths:**

FLAIR dataset utilizes multi-source data (very high resolution aerial images, elevation and Sentinel-2 time series) to solve heterogeneous spatial, temporal, and spectral resolutions domain shift in Country-scale land cover semantic segmentation, which has important application prospects in the field of multi-platform Earth observation.

**Additional Feedback:**

No.

**Correctness:**

Yes, but the design of the framework is not well-founded, and the experiments show fewer results to illustrate the characteristics and challenges of the FLAIR dataset.

**Documentation:**

Yes.

**Ethics:**

No.

**Limitations:**

The motivation for the dataset in the paper is not clear, and the design of the baseline model architecture lacks basis.

**Opportunities For Improvement:**

1. line31-line37, the motivation for utilizing multi sources optical imagery is not very clear in Section 1. The problems with existing land cover semantic segmentation datasets or a single source optical imagery are not be clearly described.
2. Figure 3, It is necessary to visualize the time series change curve of each category with different spectral band of Sentienl-2, which can help reviewers to understand the spatio-temporal distribution of each category more intuitively.
3. line100-line101, why choose 40×40 size of Sentienl-2 patches, according to the conversion of the relationship between the resolution, the size of 10×10 is enough.
4. line114-line117, the multimodality of FLAIR is not accurate. It can be thought as multiple sensor platforms. Because the imageries are all collected from optical satellite sensor.
5. line118-line123, due to the close distance between parts of the training, validation, and test sets at the time of data partitioning, similarly distributed data actually appears in both the training and validation sets. This is paradoxical for spatial domain offsets.
6. Figure 4, it is recommended to add training, validation, and test data distribution displays such as mean and standard deviation to measure the extent of different domain biases.
7. line145-line150, please describe the main reasons for metadata encoding, such as Fourier embedding, one-hot-encoded and MLP architecture.
8. line158- line 161, the “Collapsed” module is similar to the time self-attention feature fusion in U-TAE. You can consider using transformer instead of MLP in the “Collapsed” module.
9. It is recommended to set the loss weight because the semantic masks are labeled based on high spatial resolution images and adjust the weight of Laerial.
10. Table 3 and Table 4. The results of U-T&T in Table 4 are unable to be matched with the results of Table 3. Please reconfirm.
11. Figure 6, it is suggested to add images, visualization examples are too few in this paper and the supplements.


**Relation To Prior Work:**

Yes.

**Summary And Contributions:**

This paper presents a Country-Scale Land Cover Semantic Segmentation Dataset named FLAIR, which includes high-resolution aerial imagery with a ground sample distance of 20 cm and satellite time series with 10 m resolution. Compared with the previous work, the imagery resolution and meta-information attributes are more complete. In addition, a new baseline for multi-source optical imagery land cover semantic segmentation was designed to capture both detailed textures from the aerial images and complex temporal dynamics from the time series, which alleviates the challenges of spectral texture feature confusion among vegetation categories.

---

> ### Author Response · Authors · 2023-08-10
> **Response to Reviewer  1/2**
>
> We thank the reviewer for their thorough feedback and insightful comments.
>
> > the motivation for utilizing multi sources optical imagery is not very clear in Section 1.
>
> We appreciate your question about the motivation for using Sentinel-2 images in addition to VHR aerial images. This is indeed an important aspect of our research and is discussed in Section 3.2. To summarize, we've found that Sentinel-2 images offer three primary advantages over aerial images:
>  - **Increased spectral resolution:** Sentinel-2 is equipped with a 10-band multispectral imager that includes bands covering the near-infrared spectrum. This is particularly advantageous for analyzing vegetation phenology [1]. Conversely, aerial acquisitions generally contain only four spectral bands, with only one in the infrared.
> - **Multi-temporal resolution:** Sentinel-2 provides a yearly time series, allowing our model to analyze the temporal evolution of each pixel's spectral response. This is particularly useful in distinguishing between plant species, as illustrated in Figure 2 we added to the revised manuscript. In contrast, aerial acquisitions typically provide only single-date images.
> - **Larger spatial context:** The spatial resolution of Sentinel-2 is lower than that of aerial images (10m vs. 20cm). This difference, paradoxically, offers an advantage. Our model can benefit from wider receptive fields due to the larger context provided by Sentinel-2. That's why we associate each 102x102m aerial patch with a Sentinel-2 image time series covering a 400x400m extent.
> - **Spectral Consistency:** The high quality of the spectral calibration of the Sentinel-2 time series can be helpful to mitigate the shift in the radiometry induced by the radiometric corrections process of  BD Ortho.
>
> We acknowledge that these points could have been highlighted more clearly in our initial submission. Therefore, we will ensure to make these arguments more explicit in the introduction in our revised manuscript.
>
> > It is necessary to visualize the time series change curve of each category
>
> Thank you for this suggestion. We agree that the visualization of the temporal evolution of some categories using different spectral bands of Sentinel-2 would help in communicating the advantage of using this sensor. We have included a new visualization in Figure 2 of the revised manuscript.
>
> > why choose 40×40 size of Sentienl-2 patches?
>
> We selected a broader spatial context of 400 × 400m (equivalent to an image of 40x40 with 10m per pixel) to allow for the leveraging of a large spatial context. Our current baseline may not leverage the full advantage of such a large context, but our dataset allows for future methods to leverage this data. Our dataloader allows us to increase this scope up to 112x112 sentinel-2 pixels for an even larger context.
>
> > please describe the main reasons for metadata encoding
>
> We've incorporated metadata encoding to capture additional information which might be helpful in model performance. Here's a breakdown:
> - **One-hot encoding:** This method was chosen for categorical variables like acquisition year and camera type. One-hot encoding is a common and straightforward approach to transforming categorical data into a format for machine learning algorithms. By representing each category as a unique binary vector, we can prevent models from misinterpreting ordinal relationships between categories that aren't there.
> - **Fourier encoding:** These are commonly used to represent continuous variables like latitude and longitude [2] or positions in a sequence  [3,4]. Taking only low spatial frequencies prevents the network from overfitting the train set.
> - **MLP architecture:** This neural network structure was employed as a simple mechanism to map vectors of the handcrafted features to a more complex representation. MLPs can capture non-linear relationships in the data and are commonly used for tasks that involve mapping input vectors to a set of suitable outputs.
>
> [1] Misra, G., Cawkwell, F., & Wingler, A. (2020). Status of phenological research using Sentinel-2 data: A review. Remote Sensing.
>
> [2] Luc Baudoux, Jordi Inglada, and Clement Mallet. (2021) Toward a yearly country-scale CORINE land-cover map without using images: A map translation approach. Remote Sensing
>
> [3] Ashish Vaswani, Noam Shazeer, Niki Parmar, Jakob Uszkoreit, Llion Jones, Aidan N Gomez, Łukasz Kaiser, and Illia Polosukhin. (2017) Attention is all you need. In Advances in neural information processing systems,, 2017
>
> [4] Garnot, V. S. F., Landrieu, L., Giordano, S., & Chehata, N. (2020). Satellite image time series classification with pixel-set encoders and temporal self-attention. In CVPR

---

> > ### Author Response · Authors · 2023-08-10
> > **Response to Reviewer  2/2**
> >
> >
> > > the multimodality of FLAIR is not accurate.
> >
> > We appreciate your point regarding the use of the term "multimodality" for FLAIR. Indeed, both sensors are optical, albeit with widely different spatial, temporal, and spectral resolutions. To avoid potential confusion, we will revise the terminology from "multi-modal" to "multi-sensor", which more accurately describes the use of data from multiple sensor platforms.
> >
> > > add training, validation, and test data distribution displays such as mean and standard deviation
> >
> >   We appreciate the recommendation to provide a more detailed breakdown in Figure 4. However, due to the varied repartition and diverse regions encompassed within the train, test, and validation sets, presenting the mean and standard deviation of the spectral bands by dataset may not give a conclusive insight into domain biases.
> >
> > > consider using transformer instead of MLP in the “Collapsed” module.
> >
> > Our intention was to provide a foundational and straightforward approach for the "Collapsed" module, which is why we initially opted for the MLP architecture. Using transformers instead may indeed improve the results. This indeed opens up an avenue for further exploration and might pave the way for more sophisticated implementations in subsequent work.
> >
> > > set the loss weight because the semantic masks are labeled based on high spatial resolution images.
> >
> > Thank you for the recommendation. The losses Laerial and Lsat have been normalized based on the pixel count of their corresponding sensors, which allows for a straightforward summation without introducing biases. However, fine-tuning the weights can be an avenue for future exploration to see if there's a potential improvement in the model's performance.
> >
> > > Results in Table 3 and Table 4. are different
> >
> > Table 3 presents the average mIoU over 5 different runs, capturing the inherent variability with standard deviations. On the other hand, Table 4 showcases the per-class score from the best-performing run. We believe that presenting both these metrics offers distinct insights. The average IoU provides a robust estimate of the impact of different enhancement strategies. The best-performing run offers an example of performance in the single-run setting, which is the one we opted for in the competition, and it aligns with the typical reporting conventions in the literature.
> >
> >  We'll ensure to make this distinction clear in the manuscript to avoid any confusion for the readers.
> >
> > > it is suggested to add images, visualization examples are too few in this paper and the supplements.
> >
> > We've included additional spectral dynamics illustrations (Fig 2) for varied land cover classes to provide more clarity for readers unfamiliar with satellite data. However, due to page constraints, we had to be selective in our additions  We direct readers seeking more detailed visualizations to the supplementary material, where they can find more illustrations.
> >
> >  > The motivation for the dataset in the paper is not clear,
> >
> > We understand the concern and aim to provide further clarity on the motivation behind our dataset. FLAIR represents the first large-scale multi-sensor dataset with very high-resolution annotations. This unmatched resolution opens-up new geospatial analysis applications. The primary motivation is monitoring soil artificialization, a pervasive and nearly irreversible phenomenon with dire environmental effects. Beyond this, the multi-source nature of FLAIR, coupled with the sheer volume of data and high-quality annotations, makes it as an ideal resource for exploring diverse or evaluating self-, un-, or semi-supervised machine learning methods. Finally, we hope to contribute to making geospatial analysis a standard application of computer vision.
> >
> > > the design of the baseline model architecture lacks basis.
> >
> > The primary objective of this work is to introduce a novel dataset, aiming to serve as a benchmark for the research community. The baseline models were designed to offer a straightforward reference point rather than represent novel architectures. By leveraging and combining existing techniques, we intended to provide a context for future research endeavours using our dataset. We anticipate that more sophisticated and tailored approaches will yield improved results on this dataset, and we welcome the community's efforts in this direction.
> >
> > We hope the revisions and clarifications addressed the reviewer's questions and again extend our gratitude for the valuable feedback, which unequivocally improved our submission's quality.

---

> > > ### Comment · Reviewer_j7rF · 2023-08-30
> > > **Respone to rebuttal**
> > >
> > > Thank you for your reply.
> > >
> > > For the benchmark design, although the aim of this paper is to propose a novel dataset, the authors have not provided an informed benchmark design to validate the dataset. The dataset aims to fuse two sensors for land cover classification. Specifically, a dataset with multiple methods for extensive experimental validation will encourage the use of the open source community. In this study, it is insufficient for experimental validation on only U-Net and U-TAE. There is a lack of other semantic segmentation networks for high-resolution imagery and satellite imagery time series. For high-resolution semantic segmentation network, you can refer the OpenEarthMap[1] and LoveDA[2], and follow their benchmark and methods. For satellite imager time series network, you can  refer the PASITS[3], and follow their baseline and methods, such as ConvGRU,ConvLSTM,U-BiConvLSTM, U-COnvLSTM,FPN-COnvLSTM,3D-UNet. And they are open-source, it is not very difficult to extend to your benchmark.
> > >
> > > [1] J. Xia, N. Yokoya, B. Adriano, and C. Broni-Bediako, “Openearthmap: A benchmark dataset for global high-resolution land cover mapping,” in Proceedings of the IEEE/CVF Winter Conference on Applications of Computer Vision, pp. 6254–6264, 2023.
> > >
> > > [2]J. Wang, Z. Zheng, A. Ma, X. Lu, and Y. Zhong, “Loveda: A remote sensing land-cover dataset for domain adaptive semantic segmentation,” in Proceedings of the Neural Information Processing Systems Track on Datasets and Benchmarks (J. Vanschoren and S. Yeung, eds.), vol. 1, Curran Associates, Inc., 2021.
> > >
> > > [3] V. S. F. Garnot and L. Landrieu, “Panoptic segmentation of satellite image time series with convolutional temporal attention networks,” in Proceedings of the IEEE/CVF International Conference on Computer Vision, pp. 4872–4881, 2021.
> > >
> > > For the benchmark design details, it is necessary to discuss the architectural design details, such as the loss weight, the module of multi-sensor feature fusion. But the authors do not consider them. A benchmark with different experimental results and discussion will encourage more researchers to follow your datasets.

---

> > > > ### Author Response · Authors · 2023-08-30
> > > > **Response from Authors**
> > > >
> > > > We thank the reviewers for the valuable follow-up. We fully recognize the significance of benchmark design in validating our proposed dataset.
> > > >
> > > > - **Dataset's Objective:** Our primary aim is to introduce a unique, large-scale, and curated dataset to motivate the vision community to explore this specific task. We have evaluated and combined established state-of-the-art approaches for each module to serve as a foundational reference and exemplify how to use our dataloader efficiently to facilitate future work.
> > > > - **Extension to other Spatial Backbones:** In the revised manuscript, specifically within Table 5, we have added comparisons involving the FPN and DeepLab backbones for both aerial and multisensor networks.
> > > > - **On UTAE:** We are grateful for the reference to [3] and the associated methods. The superior efficiency of UTAE (ICCV21) over methods like ConvGRU (ICLR16), ConvLSTM (NeurIPS-Workshop18), U-ConvLSTM (CVPR-Workshop19), U-BiConvLSTM (IJPRS21),
> > > > FPN-ConvLSTM (IJPRS21), and 3D-UNet (CVPR-Workshop19), in terms of both performance and memory usage, is well-documented for image time series analysis. See [4, Table 2] for a meta-study on the superiority of the TAE for time series encoding over other approaches. We believe that incorporating modules already recognized as less effective by current research might distract from our primary message.
> > > > - **Real-world Implementation:** Our dataset is actively employed for the FLAIR#2 challenge ([link](https://codalab.lisn.upsaclay.fr/competitions/13447)). The challenge has already attracted significant attention, with over 100 registrations and numerous submissions. By its end on September 25, a wide array of methodologies would have undergone evaluation using our dataset, and the top-performing methods will be documented in a public technical report.
> > > > - **Implementation Details:** Details on the multi-sensor feature fusion modules can be found in the supplementary materials. For the sake of simplicity, and as highlighted in Section 4, the final loss does not incorporate weights (lambda = 1 for both modules). Our trials indicated a marginal impact of this parameter, and we aimed to prevent any over-optimization in the presentation of the benchmark.
> > > >
> > > > [4] Kondmann, Lukas, et al. "DENETHOR: The DynamicEarthNET dataset for Harmonized, inter-Operable, analysis-Ready, daily crop monitoring from space." NeurIPS Datasets and Benchmarks Track 2021.

---

> > ### Comment · Reviewer_j7rF · 2023-08-30
> > **Respone to rebuttal**
> >
> > Thank you for your reply. There are some concerns about metadata encoding, and the author didn't fully understand my question.
> >
> > For metadata encoding, the one-hot encoding is mostly used for the classification task to process the label. However, the acquisition year belongs to temporal value. The Time Embedding in transformer is more suitbable to process it. It is unfounded. Fourier encoding is the most used method to encode metadata in the field of computer vision, especially in the image generation.But the metadata involves location information such as latitude and longitude, which seems to be better suited to using the Positional Encoding. In addition, due to the different types of metadata, it is necessary to discuss whether all metadata is helpful for the landcover classification, more specifically, it is possible that not all metadata will improve the accuracy, and the importance of each type of metadata for improving classification results needs to be discussed.

---

> > > ### Author Response · Authors · 2023-08-30
> > > **Follow up**
> > >
> > > We thank the reviewer for bringing up metadata encoding.
> > >
> > > - **Acquisition Year Encoding:** We understand the reviewer's perspective on treating acquisition years as temporal values. In many contexts, temporal values are indeed processed using Fourier embeddings in transformers; we use this technique to encode the position of satellite images *within a year*. However, in our dataset, the year of acquisition has a discrete nature: each time series only spans a specific year among 2018, 2019, 2020, and 2021, *without overlap*. Consequently, the year of acquisition serves as a discrete qualifier for each time series. Implementing Fourier encoding for such a small set of discrete categories could lead to unnecessary complexities. Our choice of one-hot encoding for the acquisition years aims to facilitate the network's ability to condition the time series on year-specific latent variables, such as the rainfall or temperature patterns of that particular year.
> > >
> > > - **Longitude and Latitude Encoding:** As stated in Line 159 of our manuscript, we employ Fourier features to encode longitude and latitude values, considering them as positional encodings.
> > >
> > > - **Modularity of the Public Code:** We wish to highlight the flexibility inherent in our code base. Our publicly accessible implementation is highly modular, granting users the liberty to activate, deactivate, or customize the encoding for any given metadata according to their research requirements.

---

### Official Review · Reviewer_GDnf · 2023-07-23
**A country-wide dataset for generating land cover maps using multi-resolution aerial and satellite imagery**

**Rating:** 9
**Confidence:** 4

**Strengths:**

- Large and well organized dataset
- Good quality very high resolution land cover labels
- Challenging task to develop models able to learn from multi-resolution and multi-temporal data

**Additional Feedback:**

I hope that my comments and suggestion could help the authors to improve the manuscript.

**Clarity:**

The paper is well organized and clear. However, I have some suggestions for improving the clarity of the manuscript:

* Annotations: In the main text, it would be beneficial to briefly explain the transition from 18 classes during labeling to 12 classes. This will make it easier for the reader to understand the change without having to refer to the appendix.

* Elevation: The description and processing of the elevation information should be more explicit. It should be clarified whether the elevation data is a Digital Surface Model obtained solely through photogrammetric processing of aerial images and its level of accuracy. Additionally, if one of the stereo images used is the provided one, it should be highlighted that the DSM is a derived product from the image, not an independent measurement, and include these details in the main text.

* Very High Resolution Aerial Images: In the main manuscript, specify that orthorectified aerial imagery was used.

* Sentinel-2 Time Series: Clearly report the level of preprocessing of S2 images in the main text, not just in the appendix. Also, mention that the 20m bands were upsampled to 10m. State whether the cloud mask and snow mask are obtained from Sen2Cor.  I suggest to consider the more reliable cloud detector algorithm: https://github.com/sentinel-hub/sentinel2-cloud-detector

* Dataset Description: In the dataset description, clarify that the S2 patches cover larger areas compared to the aerial imagery.

By incorporating these clarifications, the manuscript's clarity and comprehensibility can be significantly improved.

**Correctness:**

There are no major issues with the manuscript and dataset. However, I have some minor concerns:

* Multi-modality: The term "multi-modal" might not be entirely accurate for the dataset. Since both aerial VHR and Sentinel-2 are optical sensors, the main difference lies in their resolutions (spatial, temporal, and spectral). To better describe it, the dataset could be referred to as "multi-sensor" and "multi-resolution." Moreover, the elevation data is coming from the (same) optical images and it’s not an independent modality.

* Covariate-shift: The mention of covariate-shift may not be suitable for the context of the dataset, as the goal is not to predict land cover maps at different times of the year. Instead, the model uses a yearly Sentinel-2 time series and one epoch of VHR data to estimate a yearly (mean) land cover map. I suggest to remove this aspect from the text (line 121-123).

**Documentation:**

The documentation is well done.

**Ethics:**

No unreported and relevant ethical issues.

**Limitations:**

The authors have reported the main limitation of the datasets. However, I shared some additional reflections:

- The presented baseline seems to marginally improve the results using the time series information from Sentinel-2 data. This could be intrinsically related on how the label have been generated. The annotators have use the VHR images as reference to recognize  the different land covers.
- The dataset is acquired only in France and I am not sure that we can use for investigating a domani-shit
- The dimension of the S2 patches is an arbitrary choice (maybe driven by the model architecure used as baseline)

**Opportunities For Improvement:**

* The authors should have checked the quality of the produced labels and controlled the bias induced by different annotators, establishing cross-validation checks for a small percentage of the patches.
* It would be interesting to assess the contribution of the elevation layer by comparing the accuracy of the baseline model with and without the elevation band.
* The outcomes of the data challenge and the improvements compared to the baseline model are worth examining.
* The metadata encoding experiments (both discussion and results) might not be essential to include in the main text. The considered features, such as acquisition year, camera types, altitude from sea level, and variation of latitude/longitude, may not significantly help the model due to limited variability and relevance in a regional dataset like France. For example:
    - the year of the acquisitions has only three possible values and it’s not related to the land cover type. I guess a similar situation is also true for the camera types
    - The altitude from sea level can also be derived from the elevation band
    - The variation of latitude/longitude is not really useful to explain the different in term of land covers in a region like France. It could be more relevant in case of a global dataset

**Relation To Prior Work:**

The manuscript correctly report the prior work. It is worth to notice that the FLAIR dataset is an improvement respect to the first version. The authors achieved a more interesting dataset design including the Sentinel-2 data (multi-sensor/multi-resolution)

**Summary And Contributions:**

The authors introduce the second version of the FLAIR dataset with the primary objective of generating high-resolution land cover maps. The dataset integrates VHR aerial imagery, Elevation data, and Sentinel-2 time series, creating a diverse set of data with varying spatial, spectral, and temporal resolutions, covering over 817 km2 of France's landscape.
The authors' commendable efforts in preparing accurate reference labels for precise land cover classification are evident. This dataset holds significant potential for various tests and represents an important contribution to both the remote sensing and deep learning communities.

---

> ### Author Response · Authors · 2023-08-10
>
> We thank the reviewer for their thorough feedback and insightful comments, as well as for their appreciation of our work.
>
> > The authors should have checked the quality of the produced labels
>
> Our accuracy score determination involved a manual, independent verification of around 37k randomly selected polygons, hidden from the annotating teams. The number of polygons to validate the annotations has been chosen based on a hypothesis testing approach to achieve a 95% confidence level with a 5% margin of error. The sampled area covers a total area of 18.7 km², equivalent to approximately 0.75% of the entire dataset, or 468 million pixels. If any batch of annotations did not meet the 95% accuracy criterion, it was rejected and returned for re-annotation. This iterative process fostered productive exchanges between the annotators and the geography experts from IGN, thereby ensuring a high-quality dataset.
>
> We have added this discussion in Section 6.
>
> > It would be interesting to assess the contribution of the elevation layer by comparing the accuracy of the baseline model with and without the elevation band.
>
> To assess the contribution of the elevation layer, we trained a model excluding this band and were not able to observe a discernible positive impact. We think that a more specialized architecture might be better suited to this non-radiometric band, such as a FuseNet-like approach with split encoders [1]. Nevertheless, the primary intent of our benchmark dataset is to furnish straightforward baselines that can assist subsequent research in contextualizing their results.
>
> > The metadata encoding experiments might not be essential to include in the main text.
>
> We appreciate the reviewer's perspective on the metadata encoding experiments. While our baseline may not have exploited the full potential of the metadata, it's crucial to recognize that such metadata often remains absent in Earth observation imagery datasets. We believe that these data can make a substantial difference if utilized aptly. Let's address each point:
> - **Acquisition Year:** It's accurate that the dataset comprises imagery from only a few distinct years. However, every year exhibits specific meteorological patterns, profoundly influencing plant growth. This aspect becomes crucial when contextualizing Sentinel-2 time series [2].
> - **Elevation Layer:** The elevation layer we've incorporated is derived as DEM = DSM - DTM. As a result, it's impossible to retrieve the altitude from the sea directly from this layer. The altitude, in many cases, becomes an essential parameter to distinguish between certain tree species and architectural forms.
> - **Latitude/Longitude Variation:** France is renowned for its varied bioclimatic zones, ranging from Mediterranean and continental to oceanic and mountainous climates. These differences dramatically influence the appearance and characteristics of land covers like agricultural fields or forests.
>
> In conclusion, we've added these metadata elements not just for the sake of complexity but because we believe they are often dismissed in existing datasets while they could provide beneficial insights for land cover classifications, especially when treated with appropriate techniques or architectures.
>
> > The annotators [only] use the VHR images [and not the Sentinel-2 time series]
>
> Indeed, our annotators did not use the time series from the Sentinel-2 data when annotating the aerial images. Visualizing medium-resolution (10m) time series of multispectral images can be particularly challenging for human annotators. VHR images, on the other hand, are generally viewed by experts as a reliable and sufficient reference to identify different land covers. For instance, features like the IRC band can be instrumental in differentiating vegetation types. Furthermore, it's pertinent to note that annotators had access exposed to a broader context beyond just the 512x512 patches.
>
> [1] FuseNet Hazirbas, C., Ma, L., Domokos, C., & Cremers, D. (2017). Fusenet: Incorporating depth into semantic segmentation via fusion-based CNN architecture. In ACCV.
>
> [2] Quinton, F., & Landrieu, L. (2021). Crop Rotation Modeling for Deep Learning-Based Parcel Classification from Satellite Time Series, Remote Sensing.

---

> > ### Author Response · Authors · 2023-08-10
> >
> >
> > > The dataset is acquired only in France
> >
> > Thank you for highlighting this aspect. While FLAIR focuses on French regions, it's important to clarify a few points regarding its transferability:
> > - **Source of Data**: As a national agency, IGN restricts its surveying and annotation activities within the boundaries of France. Consequently, it's challenging for us to evaluate the performance of models trained on FLAIR when applied to completely uncharted domains, especially when considering the consistency of annotation nomenclature and methodology.
> > - **Transfer Learning:** FLAIR can serve as a robust foundation for future works by using our pre-trained models and fine-tuning using data from diverse geographical locations. This is, however, out of the scope of this study.
> > - **Future Potential:** We're exploring the possibility of annotating data from France's overseas territories. These regions have distinct climates compared to metropolitan France and can provide insights into the adaptability of FLAIR-based models in varied environments. However, this exploration exceeds the current benchmark objectives set out in our this study.
> >
> > > The dimension of the S2 patches is an arbitrary choice
> >
> > The Sentinel-2 imagery's 40x40 resolution corresponds to a spatial coverage of 400m x 400m, which centers around the very high-resolution (VHR) aerial patch measuring 512x512 pixels or 102.4m x 102.4m. This configuration ensures that the Sentinel-2 patches provide a broader contextual area relative to the VHR patches. If researchers require larger context, our data loader can be easily modified to extract larger Sentinel-2 patches, up to a size of 112x112, which equates to an area of over 1.25km².
> >
> >
> >
> > >  The term "multi-modal" might not be entirely accurate for the dataset
> >
> > We appreciate your point regarding the use of the term "multimodality" for FLAIR. Indeed, both sensors are optical, albeit with widely different spatial, temporal, and spectral resolutions. To avoid potential confusion, we will revise the terminology from "multi-modal" to "multi-sensor", which more accurately describes the use of data from multiple sensor platforms.
> >
> > > Covariate-shift: The mention of covariate-shift may not be suitable for the context of the dataset
> >
> > Thank you for pointing out this aspect. We understand your perspective on the mention of covariate-shift in the context of our dataset. We removed it from the revised manuscript to ensure that the text accurately reflects the dataset's goals and methodology
> >
> > > briefly explain the transition from 18 classes during labeling to 12 classes [in the main text]
> >
> > Thank you for the suggestion. We'll provide clarification in the main text regarding this transition. Specifically, we'll include:
> >
> > "During the annotation process, we initially identified 18 classes. However, we grouped them together due to the rarity of certain classes like swimming pool, greenhouse, snow, clear-cut, ligneous, and mixed. This grouping ensures more statistically significant metrics for the evaluation. Nonetheless, users can still utilize the extended nomenclature."
> >
> > > Elevation: The description and processing of the elevation information should be more explicit.
> >
> > In section 3.2, we stated the following further clarify that the DSM is indeed derived from aerial images using photogrammetric processing, and not an independent measurement [3]. We acknowledge the importance of distinguishing between original and derived datasets and will ensure this distinction is transparently communicated in the revised manuscript.
> >
> > [3] Garioud, A., Peillet, S., Bookjans, E., Giordano, S., & Wattrelos, B. (2022). FLAIR: French Land cover from Aerospace ImageRy.
> >
> > >  specify that orthorectified aerial imagery was used.
> >
> > The 20cm high-resolution data is sourced from IGN’s "BD ortho" product, which is openly accessible and free of licence. We made this clear in the revised manuscript.
> >
> > > State whether the cloud mask and snow mask are obtained from Sen2Cor. I suggest to consider the more reliable cloud detector algorithm: https://github.com/sentinel-hub/sentinel2-cloud-detector
> >
> > In the FLAIR dataset, the preprocessing of S2 images, including the use of Sen2Cor for cloud masking, is elaborated in appendix A1. We'll shift this information to the main text for better accessibility. While we have utilized Sen2Cor for cloud masks, we recognize the sentinel2-cloud-detector as a reliable alternative and appreciate the suggestion. Including it or similar algorithms can potentially enhance the dataset's quality. Overall, FLAIR offers an excellent platform for comparing the efficiency of various cloud filtering methods, and especially their impact on deep models for satellite time series.
> >
> > We hope the revisions and clarifications addressed the reviewer's questions and again extend our gratitude for the valuable feedback, which unequivocally improved our submission's quality.

---

> > ### Comment · Reviewer_GDnf · 2023-08-29
> > **Final remarks**
> >
> > I would like to express my appreciation to the authors for carefully addressing all of my suggestions in the manuscript revision. As previously, I endorse its acceptance, considering that the dataset is well-designed and holds significant potential for widespread utilization.
> >
> > I have one final minor suggestion that can be easily fixed. You mentioned:
> >
> > > Elevation Layer: The elevation layer we've incorporated is derived as DEM = DSM - DTM. As a result, it's impossible to retrieve the altitude from the sea directly from this layer. The altitude, in many cases, becomes an essential parameter to distinguish between certain tree species and architectural forms.
> >
> > I believe that using the term "DEM" might not be accurate and could potentially cause confusion (I was confused). It's more appropriate and commonly accepted to refer to the (DSM-DTM) layer as nDSM, which stands for Normalized Digital Surface Model.

---

> > > ### Author Response · Authors · 2023-08-29
> > > **Thank you**
> > >
> > > We thank the reviewer for their appreciation of our work and for this suggested correction.
> > >
> > > We will indeed change DSM to nDSM in the next revision of the paper to avoid any misunderstanding.

---

### Official Review · Reviewer_uZv1 · 2023-07-23
**Paper Exceeds 9 Page Limit**

**Rating:** 1
**Confidence:** 5
**Clarity:** Yes

**Strengths:**

Significance: Most remote sensing datasets are of very low resolution from public satellite (10m resolution) or extremely high but narrowly defined datasets (e.g. drones of 1-2 fields at 2cm).  As satellite capabilities are rapidly improving and is becoming available at 5-20cm, this data is particularly interesting.

Relevance: Remote sensing is an important area within the broader research community, but is also an additionally important domain to study as most algorithms are currently designed for natural scenes imagery and the challenges of remote sensing are often ignored.  The dataset captures key issues when working with remote sensing such as multiple sensors, domain shift, and imbalanced label classes.

Quality: The annotations were generated by geography experts at IGN.  The baseline methods based on U-Net are reasonable.  Models are trained 5 times so that standard deviations can be captured.

**Additional Feedback:**

CRITICAL: The paper exceeds the 9 page limit and therefore cannot be accepted in its current form.  Section 7 appears on page 10 of the submission.   Even if this is reduced during camera ready, I don't believe the paper can be accepted because that would be unfair to other authors who have made the effort to reach the appropriate length.  Nevertheless, I have provided feedback here to help the authors in their future revisions.

**Correctness:**

The dataset is constructed well, but detailed descriptions are missing from the body of the text (currently located in the appendix).

The benchmarking performed is reasonable.  The analysis shows results on different combinations of input aerial and aerial+satellite.  It would also be useful to show satellite-only as this is the data that is available to the remote sensing community broadly.  If it's true that training on this small amount of high resolution imagery and labels would help the low resolution analysis when that other imagery is not available, that would significantly increase the impact of the work.

**Documentation:**

No.  Minimal information is given in the body of the text.  The authors state in the submission portal " The full dataset is currently only available after subscribing to the challenge on codalab. All data and labels will be released at the end of the challenge in September."  This prevents us from reviewing the dataset fully.  Maintenance and further development is not discussed.

**Limitations:**

The authors discuss the primary limitation which is that this dataset is limited to French regions and therefore lacks some of the geography and vegetative diversity that would be found in other regions.  It would be useful to quantify how well this data transfers to an entirely unseen domain.

The baselines rely on the 20cm high resolution data, which has not been discussed as to how it was collected.  Therefore it is unclear how this could be extended to other areas.

**Opportunities For Improvement:**


Quality: Details about the annotation and data acquisition process should be included in the body of the paper and not the appendix.  As a dataset submission, this is the heart of the paper.  From the main paper, it is unclear how the 20cm data is obtained, what the sensor parameters are, or what preprocessing occurs (beyond orthorectification).  It is mentioned that multiple cameras with different characteristics were used, but this is largely not addressed.  Additionally, the expert annotators are referenced, but efforts around QA (how many people look at each image, what if there is discrepancy, etc.) is not discussed.  I believe all of this needs to be included in the main paper for acceptance.

Due to the corrections performed to ensure homogeneity, the authors state that the colors should not be interpreted as physical channel reflectance.  It would be preferable to provide the raw reflectances so that those values can be used and then provide a transform (either in code or simply described) which could be used to obtain the domain homogeneity.

Significance: The sentinel-2 imagery itself is not novel, but the curation to create this particular dataset is helpful.  40x40 are very small tiles.  Most analysis is likely to need areas much larger than this.  The baselines rely on the 20cm high resolution data, which has not been discussed as to how it was collected.  Therefore it is unclear how this could be extended to other areas.



**Relation To Prior Work:**

There is not a significant amount of related work discussed, though many are included in the first table.

**Summary And Contributions:**

The authors introduce FLAIR, a high-resolution (20cm) multimodal land-cover dataset.  It includes 19 landcover classes and covers 817km^2 over 50 French sub-regions.  916 Patches (samples) include a 20cm yearly image and 10m satellite images.  Annotations were perfromed by IGN geography experts.

---

> ### Author Response · Authors · 2023-08-10
> **Clarification on Page Limit**
>
> We sincerely appreciate the reviewer's detailed review, which raises several pertinent points. We would like first to address your concerns regarding the page limit.
>
> As stated in the [Formatting Instructions for NeurIPS 2023](https://media.neurips.cc/Conferences/NeurIPS2023/Styles/neurips_2023.pdf):
>
> > "Papers may only be up to nine pages long, including figures. **Additional pages containing only acknowledgments and references are allowed.**"
>
> Furthermore, the [Submission Instructions](https://nips.cc/Conferences/2023/CallForDatasetsBenchmarks) for the NeurIPS 2023 Datasets and Benchmarks Track mention:
>
> > "Submissions are limited to 9 content pages in NeurIPS format, including all figures and tables; **additional pages containing** the required paper checklist (included in the template), references, and **acknowledgements are allowed.**"
>
> Based on the above, we have prepared our submission so that the acknowledgements  (Section 7) do not count towards the 9-page limit. We prepared the submission in good faith; if there has been a misunderstanding, we are more than willing to make necessary corrections.
>
> Lastly, as mentioned in the Formatting Instructions:
>
> > "Papers that exceed the page limit will not be reviewed."
>
> Given that our paper has undergone review, we assume it has passed this preliminary check.
>
> We have requested the area chair's input on this matter to ensure we are all on the same page and that any misunderstandings can be clarified promptly.

---

> > ### Author Response · Authors · 2023-08-10
> > **Discussion on Feedback 1/2**
> >
> > We now address the rest of the reviewer's comments.
> >
> > > how the 20cm data has not been discussed as to how it was collected and what preprocessing occurs
> >
> > Our research relies on a finished product from IGN, specifically the BD Ortho, which encompasses the entire territory of France in open access and is under a free license. This data is not intended to represent physical quantities. As detailed on IGN’s website, the post-processing includes: radiometric corrections (enhancement, spreading) and geometric corrections of the raw images using mosaicking graphs of the aerial acquisitions; see [1] for more details. We made this clear in the revised manuscript.
> >
> > > It is mentioned that multiple cameras with different characteristics were used, but this is largely not addressed
> >
> > Thank you for pointing out the need for clarity regarding the camera types used for aerial acquisitions. Specifically, we employed two camera systems: the IGN CAMv2 System [2] and Vexcel Imaging’s UltraCam Eagle Mark3. We've now emphasized this distinction in the main text of the manuscript.
> >
> > > efforts around QA is not discussed
> >
> > Our accuracy score determination involved a manual, independent verification of around 37k randomly selected polygons, hidden from the annotating teams. The number of polygons to validate the annotations has been chosen based on a hypothesis testing approach to achieve a 95% confidence level with a 5% margin of error. The sampled area covers a total area of 18.7 km², equivalent to approximately 0.75% of the entire dataset, or 468 million pixels. If any batch of annotations did not meet the 95% accuracy criterion, it was rejected and returned for re-annotation. This iterative process fostered productive exchanges between the annotators and the geography experts from IGN, thereby ensuring a high-quality dataset.
> >
> > We have added this discussion in Section 6.
> >
> > > It would be preferable to provide the raw reflectances
> >
> > As mentioned above, we utilized the finished product from IGN, the BD Ortho, which, unlike the raw data, is both open-access and under a free license. Therefore, it is unfortunately not feasible for us to share the raw data.
> >
> > > 40x40 are very small tiles [for Sentinel-2]
> >
> > The 40x40 tile size for the Sentinel-2 imagery corresponds to a spatial coverage of 400m x 400m, which centers around the very high-resolution (VHR) aerial patch measuring 512x512 pixels or 102.4m x 102.4m. This configuration ensures that the Sentinel-2 patches provide a broader contextual area relative to the VHR patches. If researchers require a larger context, our data loader can be easily modified to extract larger Sentinel-2 patches, up to a size of 112x112, which equates to an area of over 1.25km².
> >
> > > It would be useful to quantify how well this data transfers to an entirely unseen domain [outside of France]
> >
> > A : Thank you for highlighting this aspect. While FLAIR focuses on French regions, it's important to clarify a few points regarding its transferability:
> > -  **Source of Data**: As a national agency, IGN restricts its surveying and annotation activities within the boundaries of France. Consequently, it's challenging for us to evaluate the performance of models trained on FLAIR when applied to completely uncharted domains, especially when considering the consistency of annotation nomenclature and methodology.
> > -  **Transfer Learning:** FLAIR can serve as a robust foundation for future works by using our pre-trained models and fine-tuning using data from diverse geographical locations. This is, however, out of the scope of this study.
> > -  **Future Potential:** We're exploring the possibility of annotating data from France's overseas territories. These regions have distinct climates compared to metropolitan France and can provide insights into the adaptability of FLAIR-based models in varied environments. However, this exploration exceeds the current benchmark objectives set out in our study.
> >
> > [1] Chandelier, L., & Martinoty, G. (2009). Radiometric aerial triangulation for the equalization of digital aerial images and orthoimages. Photogramm. Eng. Remote Sens, 75(2), 193-200.
> >
> > [2] Souchon, J. P., Thom, C., Meynard, C., Martin, O., & Pierrot‐Deseilligny, M. (2010). The IGN CAMv2 System. The Photogrammetric Record, 25

---

> > > ### Author Response · Authors · 2023-08-10
> > > **Discussion on Feedback 2/2**
> > >
> > > > It would also be useful to show satellite-only [baselines]
> > >
> > > The large discrepancy between the resolution of the annotation (20cm) and of Sentinel-2 images (10m) complicates the training sentinel-only methods. To gauge the feasibility, we performed an experiment where we upscaled each Sentinel-2 image to match the 20cm resolution and subsequently trained a U-TAE network. However, this approach was computationally intensive—given that each pixel now encapsulates data from 20-110 images across 10 spectral bands—and the outcomes were suboptimal, recording a modest mIoU of 35%.
> > >
> > > While our initial attempts using satellite-only images were not as successful, we believe that there's potential for sophisticated methodologies to generate improved results. We see FLAIR as a cornerstone for such future endeavours. For completeness, we will incorporate our satellite-only baseline findings in the appendix of the paper.
> > >
> > > We hope the revisions and clarifications addressed the reviewer's questions and again extend our gratitude for the valuable feedback.

---

> > > > ### Author Response · Authors · 2023-08-29
> > > > **Satellite-Only  Baseline**
> > > >
> > > > IWe have made a recent update we made to our manuscript:
> > > >
> > > > - **Satellite-Only Baseline**: We have added the performance metrics of the satellite-only baseline in Table 6 of the main document.
> > > > -  **Method Modification:** Since our last message, we have slightly altered the methodology. Rather than upsampling the input, we upsample the prediction. This tweak has resulted in an enhanced performance, now standing at 36.9%.
> > > > - **Comparative Performance:** Even with the modified approach, the performance of the satellite-only baseline remains notably below that of both the aerial and multi-sensor approaches.
> > > >
> > > > We hope that these updates provide a clearer picture of our research and its comparative benchmarks. We're eager to hear your feedback and any further suggestions you might have.

---

### Official Review · Reviewer_pqJp · 2023-07-28
**Semantic Segmentation Dataset From Multi-Source Aerial Imagery**

**Rating:** 7
**Confidence:** 4
**Correctness:** The data is constructed in a sound wa…
**Clarity:** The paper is well-written, and its co…

**Strengths:**

The paper is well-written, with clear data specifications. The dataset is solid and collected in a rigorous way. The paper also included a good comparison with popular existing data sets such as MultiSenGE, MiniFrance, OpenEarthMap and BigEarthNet.


**Additional Feedback:**

NIL

**Documentation:**

The documentation is sufficient and well-organised. It is also used for a data challenge.

**Ethics:**

NIL

**Limitations:**

The authors have adequately discussed the limitations. It is also good to consider multimodalities and dataset shifts in the dataset.

**Opportunities For Improvement:**

The dataset has resolutions of 20cm for aerial images and 10m for satellite images, which are, however, limited for remote sensing applications. It will be very useful to acquire satellite images of higher resolutions, such as 1m or 0.3m. The chosen baseline models are popular but limited, and it will be good to also provide baselines using FCN and DeepLab.

**Relation To Prior Work:**

It is clearly discussed and compared with previous work.

**Summary And Contributions:**

The paper introduced a multi-source aerial imagery dataset of over 817 km2 in France with 13 different labels. The authors have also considered the multimodalities and domain shifts over multiple seasons and metropolitan territories. The dataset is interesting and useful to the research community.

---

> ### Author Response · Authors · 2023-08-10
> **Thank you for your feedback**
>
> We thank the reviewer for their insightful comments and appreciation of our work. We have uploaded a new revised version of the paper and provide responses to each of the points mentioned.
>
> >  It will be very useful to acquire satellite images of higher resolutions, such as 1m or 0.3m.
>
> Indeed, we acknowledge the difference in spatial resolution between the aerial and satellite imagery in our dataset. While incorporating higher-resolution satellite imagery, such as 1m or 0.3m, would be valuable, it's important to note that such imagery often comes with associated costs and strict licensing restrictions. A key advantage of the Sentinel-2 time series is its superior temporal resolution, with a revisit time of only 5 days. In contrast, civilian satellite platforms that offer meter-level spatial resolution typically have longer revisit intervals, a trade-off exemplified by Sentinel satellites.
>
> > it will be good to also provide baselines using FCN and DeepLab.
>
> Thank you for the insightful suggestion. In response, we've incorporated a comparative analysis where we changed the aerial image backbone's decoder from U-Net to Feature Pyramid Network (FPN) and DeepLabV3. We conducted tests under two configurations: with and without the integration of our proposed enhancement strategies (FILT, AVG M, MTD, AUG). The outcomes of this investigation are given in Table 5 of the updated manuscript. A small performance uptick was identified when deploying the larger DeepLabV3 model.
>
> We hope the revisions and clarifications addressed the reviewer's questions and again extend our gratitude for the valuable feedback.

---

### Official Review · Reviewer_g3Jt · 2023-07-29
**A French, large-scale high-resolution urban land cover dataset for fusion with Sentinel-2**

**Rating:** 7
**Confidence:** 4

**Strengths:**

The paper is cleanly organized, written in plain English to the point, and supported by figures and tables summarizing the relevant aspects of the work.

**Additional Feedback:**

Thank you for a clean and well-organized submission.

**Clarity:**

The paper is clearly written and well structured including paragraphs on ethics and societal implications.

**Correctness:**

Construction of the dataset and conducting corresponding baseline benchmarks (intersection-over-union including error bars) are sound and appear correct.

**Documentation:**

The work ships with a demo and code repository https://github.com/IGNF/FLAIR-2-AI-Challenge under the Apache 2.0 license. The release of the full dataset is scheduled for end of Sep 2023. The dataset license complies with various open standards like the Creative Commons Attribution (CC-BY). Two data challenges associated document usability of the data and demonstrating success of community models operated on.

**Ethics:**

Sensitive terrain like nuclear power plants and military facilities are said to be excluded from the dataset.

**Limitations:**

The dataset is limited to French territory, and the Sentinel-2 multi-spectral data is not cloud-cover filtered.

**Opportunities For Improvement:**

FLAIR adds yet another geospatial dataset to high-resolution urban land cover classification such as DeepGlobe. However, it excells in size and annotation verified by human expert visual inspection. Some more details on the annotation procedure in the supplementary material would be appreciated, e.g. how is determined the accuracy score over 95% as stated in L222?

**Relation To Prior Work:**

Relations to prior art are summarized in Tab. 1 along with a paragraph of discussion in the main text.

**Summary And Contributions:**

The work presents a geospatial, .2m-resolution semantic segmentation dataset (19 classes) covering a total of about 30x30 km^2 of French urban scenes photographed in RGB-NIR channels. The set is augmented by a .2m digital elevation model derived from photogrammetry and ships with a full-year time series of Sentinel-2 imagery (not cloud filtered). The dataset is currently the base of a data science competition, https://codalab.lisn.upsaclay.fr/competitions/13447 . The authors provide code for a U-Net-like baseline model.

---

> ### Author Response · Authors · 2023-08-10
> **Thank you for the feedback**
>
> We thank the reviewer for their insightful comments and appreciation of our work. We have uploaded a new revised version of the paper and provide responses to each of the points mentioned:
>
> > how is determined the accuracy score over 95% as stated in L222?
>
> Our accuracy score determination involved a manual, independent verification of around 37k randomly selected polygons, hidden from the annotating teams. The number of polygons to validate the annotations has been chosen based on a hypothesis testing approach to achieve a 95% confidence level with a 5% margin of error. The sampled area covers a total area of 18.7 km², equivalent to approximately 0.75% of the entire dataset, or 468 million pixels. If any batch of annotations did not meet the 95% accuracy criterion, it was rejected and returned for re-annotation. This iterative process fostered productive exchanges between the annotators and the geography experts from IGN, thereby ensuring a high-quality dataset.
>
> We have added this discussion in Section 6 in the revised manuscript.
>
> > The dataset is not cloud-cover filtered.
>
> Thank you for raising the point about cloud filtering. While our dataset does include raw, unfiltered Sentinel-2 imagery, we have also incorporated cloud and snow probability masks, as detailed on line 112 of the revised manuscript. These masks enable users to implement their own cloud cover filtering techniques tailored to their specific requirements.
> We've also provided an example of a basic cloud filtering method in the dataset: as outlined on line 196, images with more than 60% of pixels displaying a cloud cover probability exceeding 50% are removed. Implementing this simple step yielded a performance improvement of 0.3%.
>
> We hope the revisions and clarifications addressed the questions of the reviewer and again extend our gratitude for the valuable feedback.

---

### Author Response · Authors · 2023-08-29
**Rebuttal**

Dear Reviewers and Area Chair,

We would like to reiterate our gratitude for your comprehensive reviews of our submission. As the discussion period draws to a close, we wanted to ensure a few things:

- That the reviewers have been notified of our rebuttal and revised manuscript, which we submitted early in the response period.

- That the area chair had the opportunity to discuss with Reviewer uZv1 about our understanding of the submission guidelines, especially concerning the page limit.

Your guidance and clarification on these matters will be greatly appreciated.

---

### Decision · Program_Chairs · 2023-09-22

**Decision:**

Accept (Poster)

**Comment:**

The paper under review presents a geospatial, high-resolution semantic segmentation dataset covering French urban scenes. Multiple reviewers have provided their feedback and suggestions, and the authors have responded to these comments and made revisions.
Reviewers have highlighted several strengths of the paper:

•	The dataset is well-organized and of high quality.

•	The paper addresses the challenge of developing models capable of learning from multi-resolution and multi-temporal data, making it significant.

•	The paper represents an important contribution to both the remote sensing and deep learning communities.

Several concerns and suggestions for improvement have been raised by reviewers, and the authors have addressed these effectively:

•	Detailed information about the annotation procedure and cloud cover filtering has been added to the manuscript.

•	The authors have provided explanations for the choice of Sentinel-2 imagery and discussed the limitations and advantages.

•	They have incorporated a comparative analysis using different models, addressed metadata encoding methods, and clarified terminology.

•	Visualizations of time series changes have been added to aid understanding.

•	The authors have explained the choice of patch size and addressed concerns about transferability outside of France.

•	Additional visualizations and clarification of motivation have been provided.

While some suggestions, such as using transformers instead of MLP in a specific module, were not fully adopted, the authors have provided reasoning for their choices. Discrepancies between tables have been clarified.

Overall, the authors have diligently addressed the comments and suggestions from the reviewers, enhancing the clarity and quality of the paper. The reviewers' concerns have been effectively resolved. Considering the positive feedback and the authors' thorough responses, the paper is well-prepared for acceptance.

PS: the criticism related to the page limit is clearly explained by the authors, thus, did not impact the decision. After removing the extremely low rank due to this point, this paper shall have an average score of 7.